# Integrated omics networks reveal the temporal signaling events of brassinosteroid response in *Arabidopsis*

Natalie M. Clark [ID] [1], Trevor M. Nolan [ID] [2,4], Ping Wang [ID] [2], Gaoyuan Song[1], Christian Montes [ID] [1], Conner T. Valentine[1], Hongqing Guo [ID] [2], Rosangela Sozzani[3], Yanhai Yin[2] & Justin W. Walley [ID] [1✉]

Brassinosteroids (BRs) are plant steroid hormones that regulate cell division and stress response. Here we use a systems biology approach to integrate multi-omic datasets and unravel the molecular signaling events of BR response in *Arabidopsis*. We profile the levels of 26,669 transcripts, 9,533 protein groups, and 26,617 phosphorylation sites from *Arabidopsis* seedlings treated with brassinolide (BL) for six different lengths of time. We then construct a network inference pipeline called Spatiotemporal Clustering and Inference of Omics Networks (SC-ION) to integrate these data. We use our network predictions to identify putative phosphorylation sites on BES1 and experimentally validate their importance. Additionally, we identify BRONTOSAURUS (BRON) as a transcription factor that regulates cell division, and we show that *BRON* expression is modulated by BR-responsive kinases and transcription factors. This work demonstrates the power of integrative network analysis applied to multi-omic data and provides fundamental insights into the molecular signaling events occurring during BR response.

[1] Department of Plant Pathology and Microbiology, Iowa State University, Ames, IA 50011, USA. [2] Department of Genetics, Developmental, and Cell Biology, Iowa State University, Ames, IA 50011, USA. [3] Department of Plant and Microbial Biology, North Carolina State University, Raleigh, NC 27695, USA. [4]Present address: Department of Biology, Duke University, Durham, NC 27708, USA. ✉email: jwalley@iastate.edu

Brassinosteroids (BRs) are involved in a number of important biological processes including cell elongation and division, photomorphogenesis, reproduction, and both biotic and abiotic stress responses. The BR-signaling pathway has been well-established in *Arabidopsis*[1–3]. BRs are first sensed by the plasma membrane-localized receptor BRASSINOSTEROID INSENSITIVE 1 (BRI1)[4–6]. Upon BR binding, BRI1 recruits co-receptors including BRI1-ASSOCIATED RECEPTOR KINASE 1 (BAK1) that are required for its activation[7,8]. In the absence of BR, the GSK3 kinase BRASSINOSTEROID INSENSITIVE 2 (BIN2) phosphorylates numerous substrates including the transcription factors (TFs) BRI1-EMS-SUPPRESSOR 1 (BES1), and BRASSINAZOLE-RESISTANT 1 (BZR1)[7,9–12]. This phosphorylation inactivates BES1 and BZR1 through cytoplasmic retention, degradation, and/or reduced DNA binding. When BRs are present, the BRI1/BAK1 complex activates a kinase-signaling cascade resulting in the inactivation of BIN2 and the dephosphorylation of BES1 and BZR1. This allows BES1 and BZR1 to regulate target gene expression in the nucleus[12–18]. While there are many conserved GSK3 phosphorylation sites in BES1 and BZR1 proteins, the exact sites that are phosphorylated by BIN2 and the sites responsible for negative regulation of BES1 activity are not fully defined[19,20].

The role of BR in modulating cell division is well-documented, particularly in the *Arabidopsis* root. It has been shown that BL has a dose-dependent effect on cell division in the root meristem: roots treated with higher levels of BL have more meristematic cells. In addition, the *bes1-D* gain-of-function and *bri1-116* loss-of-function mutants have altered cell cycle progression, which implicates that plant cyclins may be involved. Accordingly, it has been shown that CYCLIN D3;1 (CYCD3;1) is induced by BL and can rescue cell division defects in the *bri1-116* mutant[21,22]. BR has also been implicated in stem cell division and maintenance: roots treated with BL have excessive Quiescent Center (QC) divisions and altered expression patterns of QC cell identity markers[22–24].

The BR-signaling pathway in *Arabidopsis* is highly dependent on the levels of protein phosphorylation, modulation of protein levels, and downstream transcriptional regulation. Therefore, we determined the dynamic response to BR signaling in *Arabidopsis* by performing large-scale transcriptome and (phospho)proteome profiling of seedlings treated with BL for different lengths of time. We then inferred a set of integrated, TF-centered Gene Regulatory Networks (GRNs) using our newly-developed Spatiotemporal Clustering and Inference of Omics Networks (SC-ION) pipeline. These networks illustrated how the phosphorylation state of TFs is important for predicting their downstream target genes. In addition, SC-ION allowed us to infer one network per time point and visualize the early and late (phospho) protein-transcript regulations in BR response. By combining these TF-centered networks with a correlation-based kinase-signaling (i.e., kinase-centered) network, we illustrated the temporal cascade of BR response starting with kinase-signaling and ending with differential transcript abundance. We were able to use this network to predict and experimentally validate previously uncharacterized BIN2 phosphorylation sites on BES1. In addition, through network motif analysis, we identified a number of TFs putatively involved in BR response. In particular, we identified a C2H2-like TF whose mutants displayed hypersensitivity to BR, longer roots, and more cell divisions. The combination of this mutant's developmental phenotype, as well as its putative role in BR response, led us to name this TF BRONTOSAURUS (BRON). Together with our results provide a comprehensive guide to molecular signaling changes that occur in response to BR.

## Results

**Generating an integrated omics time course of BR response.** To investigate the temporal response to BRs, we established a treatment system in which seedlings were sensitized to BRs by pre-treatment with 1 μM of the BR biosynthesis inhibitor brassinazole (BRZ)[25] for 7 days to reduce background BR signaling. The 7-day-old seedlings were then treated with a mock solvent or 1 μM BL for six different lengths of time (15 min, 30 min, 1 h, 2 h, 4 h, 8 h) (Fig. 1a). To confirm the efficacy of the BL treatment, we assayed BES1 by western blot (Supplementary Figure 1). In BRZ-treated seedlings, we found that BES1 predominantly exists in its phosphorylated form, while BL-treated seedlings showed an accumulation of dephosphorylated BES1 over time. Specifically, we observed an increase in the amount of dephosphorylated BES1 as early as 15 min after treatment, and the phosphorylated form of BES1 was undetectable by 1 h after treatment (Supplementary Figure 1). This demonstrates the expected BES1 response and thus the efficacy of BL treatment.

Based on the dynamic nature of BL response, we expected many transcripts, proteins, and phosphosites would have differential responses depending on the length of BL treatment. Thus, we performed multi-omics profiling of BL-treated and mock-treated seedlings at each of the six timepoints (Fig. 1a). We used 3′ QuantSeq[26] to measure transcript levels and quantified protein abundance and phosphorylation level by performing two-dimensional liquid chromatography-tandem mass spectrometry (2D-LC-MS/MS) on Tandem Mass Tag (TMT) labeled peptides[27–31].

To facilitate the analysis of complex multi-run proteomics data sets, we constructed an analysis pipeline for quantitative proteomics data called TMT Normalization, Expression Analysis, and statistical Testing (TMT-NEAT). Our pipeline, which works on data generated from any organism, takes the TMT reporter ion intensity values (i.e., MaxQuant proteinGroups or PTM_Sites) and a metadata file containing sample information and TMT-labeling scheme as input. It first cleans the data by removing contaminants and appropriately labels the intensity data using the metadata file. Second, TMT-NEAT performs sample loading (within-run) and internal reference (between-run) normalization to eliminate batch effects[32]. Third, it provides multiple qualitative plots such as hierarchical clustering and principal component analysis to visualize differences between biological groups. Finally, it performs differential expression analysis on the normalized values using a user-supplied *p*- or *q* value threshold (Fig. 1b). TMT-NEAT is publicly available and can be run through an RShiny Graphical User Interface (GUI) (see Methods).

Using these methods, we identified 26,669 transcripts, 9533 protein groups, and 26,617 phosphosites (arising from 5865 phosphoproteins) across the six timepoints (Fig. 1c, Supplementary Data 1–3). We found that the number of differentially expressed (DE) transcripts, proteins, and phosphosites varied depending on the time point. When we examined the transcript data, we found that only 208 transcripts (17 up, 191 down) were DE in response to BL within the first 15 min, whereas 454 protein groups (214 up, 250 down) and 590 phosphosites (237 up, 353 down) were DE at this same time point. In addition, many more transcripts (2653 total: 1247 up, 1406 down) were DE beginning at 30 min, which led us to speculate that the early BR response is predominantly post-transcriptional. This is supported by the role of BES1 and BZR1, which must be dephosphorylated in order to enter the nucleus and transcriptionally regulate downstream genes in response to BR[9,12,16].

We next performed Gene Ontology (GO) analysis on the DE transcripts, proteins, and phosphosites at early (1 h or earlier)

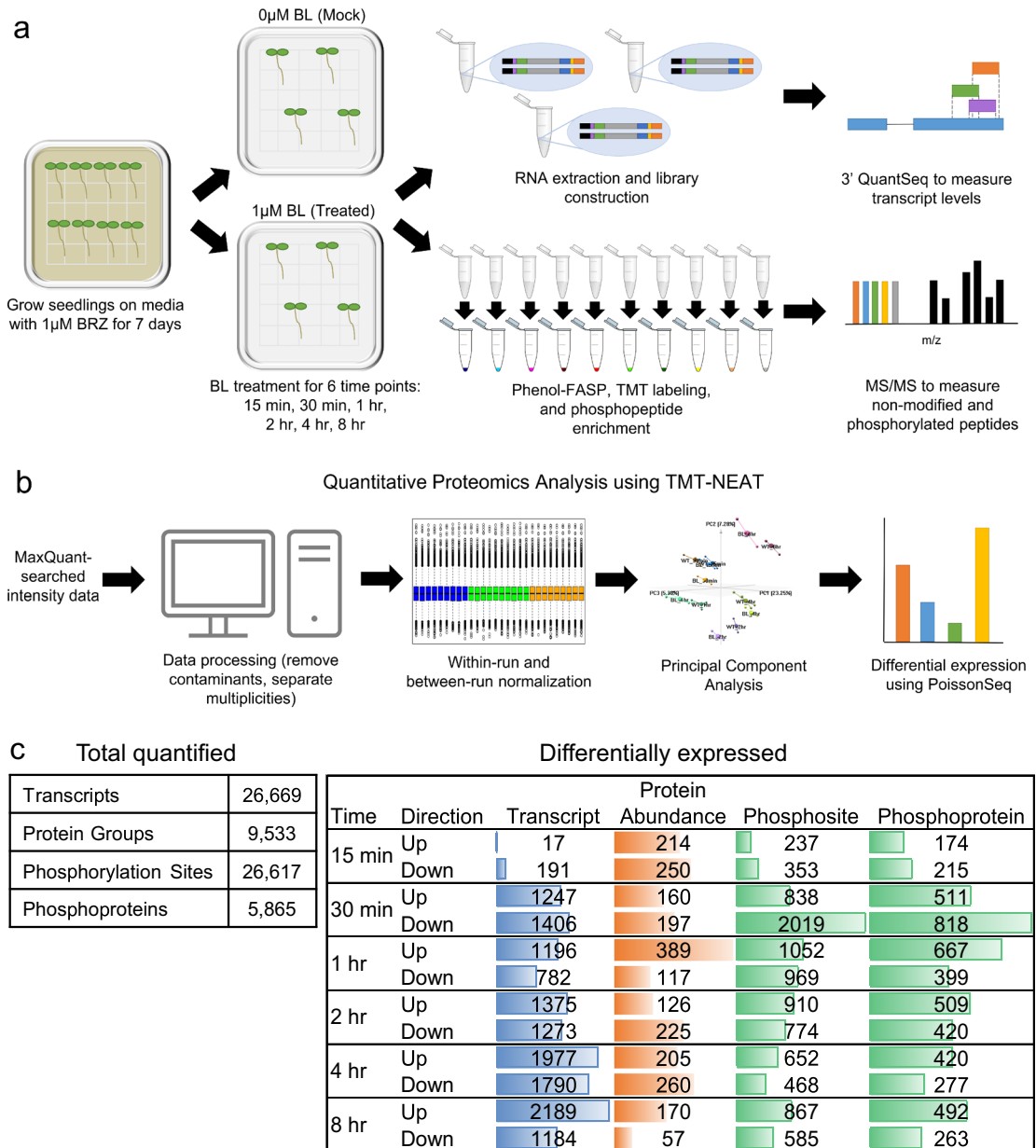

**Fig. 1 An integrated omics time course of BR response. a** Samples were collected at six timepoints (15 min, 30 min, 1 h, 2 h, 4 h, 8 h) from Mock and BL-treated seedlings. **b** TMT-NEAT analysis pipeline for quantitative proteomics. **c** (left) Total number of transcripts, proteins, and phosphosites/phosphoproteins quantified at each time point. (right) The number of DE transcript, proteins, and phosphosites/phosphoproteins at each time point. Colored bars represent the relative number of DE gene products within each data type.

and late (after 1 h) timepoints (Supplementary Figure 1, Supplementary Data 4). We found that multiple BR-response terms are enriched at different time points and for different DE gene products. For example, BR-mediated signaling pathway and cellular response to BR stimulus terms are enriched in the DE phosphosites at both early and late time points. In addition, BR biosynthetic process and BR homeostasis are enriched in the DE transcripts at early timepoints, and the term for response to brassinosteroid is enriched in both DE transcripts and phosphosites. We also see enriched terms that have been linked to BR response such as cellular response to stress, response to auxin-mediated signaling pathway, and defense response[33]. Taken together, our omics profiling captures how different gene products temporally respond to BR treatment in *Arabidopsis*.

**Predictive networks illustrate the BR temporal cascade**. We have previously shown that integrating mRNA, protein, and phosphorylation data sets greatly improves the predictive power of reconstructed GRNs[34]. However, integrating these multi-omics data types remains challenging. Thus, we next developed a network inference pipeline, which we named SC-ION, to integrate any number of different types of expression profiles into one cohesive, predictive GRN (Fig. 2a, see Methods). SC-ION builds on our MATLAB-based pipeline Regression Tree Pipeline for Spatial, Temporal, and Replicate data (RTP-STAR)[35,36], which is an adaptation of the GENIE3[37] network inference method and functions only on transcriptomic data sets. In SC-ION, we further improve on RTP-STAR by: (1) incorporating Dynamic Time Warping (DTW) clustering for temporal data[38] and Independent Component Analysis (ICA) clustering for non-temporal data[39];

**Fig. 2 Timepoint-specific, integrative omics networks of BR response. a** SC-ION pipeline. **b** Timepoint-specific networks using TF abundance (left) or TF phosphosite intensity (right). Edge color represents the time point that regulation is predicted to occur. Nodes represent individual genes. The nodes are arranged in a circular layout relative to timing—genes differentially expressed at the earliest time point (15 min) are placed at the top of the circle, and time increases as one moves clockwise through the layout.

(2) allowing the user to provide separate regulator and target matrices for integration of DE gene products; (3) integrating any number of different types of expression profiles into one GRN; and (4) providing our pipeline as an RShiny GUI (Fig. 2a).

Here, we used SC-ION to infer two separate TF-centered networks of BR response. In these TF-centered GRNs, TFs serve as "regulators" used to infer their "target" genes. In the first network, which we call the abundance network (blue, solid edges, Fig. 3), TF protein abundance (when quantified) or TF transcript abundance (when cognate protein was not quantified) was used as the "regulator" value to infer their "target" transcript abundance. In the second phosphosite network (green, dashed edges, Fig. 3), we inferred the "target" transcript abundance using TF phosphosite intensities as the "regulator" value. For each of these networks, we took advantage of our temporal data by (1) clustering the gene products using DTW to create temporally informed regulatory modules and (2) inferring one network per time point to visualize the regulations unique to each time point (Fig. 2b). When connecting the timepoint-specific networks, we found that there were distinct clusters of early and late predictions. In addition, there is cross-communication between the time points, where early regulators feed-forward into late regulators, and conversely where late regulators feedback onto early regulators (Fig. 2b).

In addition to our SC-ION-generated TF-centered GRNs, we used our correlation-based approach to infer a kinase-signaling network (purple, dotted edges, Fig. 3; see Methods)[40]. In this network, we considered kinases with DE phosphosites in their

p-loop (also termed activation loop) domain as potential regulators, as this phosphorylation in this region is necessary for kinase activity[41]. Thus, activation loop phosphorylation can be used to infer kinase activity and is useful for predicting kinase-signaling[40,42,43]. In agreement with our previous work in maize[40], we observed that the correlation between kinase protein abundance and kinase p-loop phosphorylation intensity (i.e., activation state) greatly varies depending on the time point (Supplementary Figure 2), motivating our use of p-loop site intensities rather than simply kinase abundance in our network. This "kinase-centered" network complements our TF-centered GRNs by predicting kinase-dependent signaling events.

When we merged our kinase-signaling network with our TF-centered networks, we found that genes with known roles in the BR response pathway[1] were significantly enriched in the list of network regulators (Hypergeometric test, $p < 0.001$). In addition, we noticed that this merged network illustrates the temporal cascade of BR response across these different omics levels (Fig. 3, Supplementary Data 5). Our inferred network places the kinase-signaling interactions (purple) towards the top of the network (early in time). Next are the TF phosphosite-level regulations (green), followed by the TF abundance-level regulations (blue). Thus, our network predicts that BR response begins with kinase signaling, followed by transcriptional regulation via modulation of TF phosphorylation and/or abundance. This network prediction is in agreement with what we currently know about BR signaling, which begins with the kinases BRI1 and BAK1 initiating a series of (de)phosphorylation events leading to the

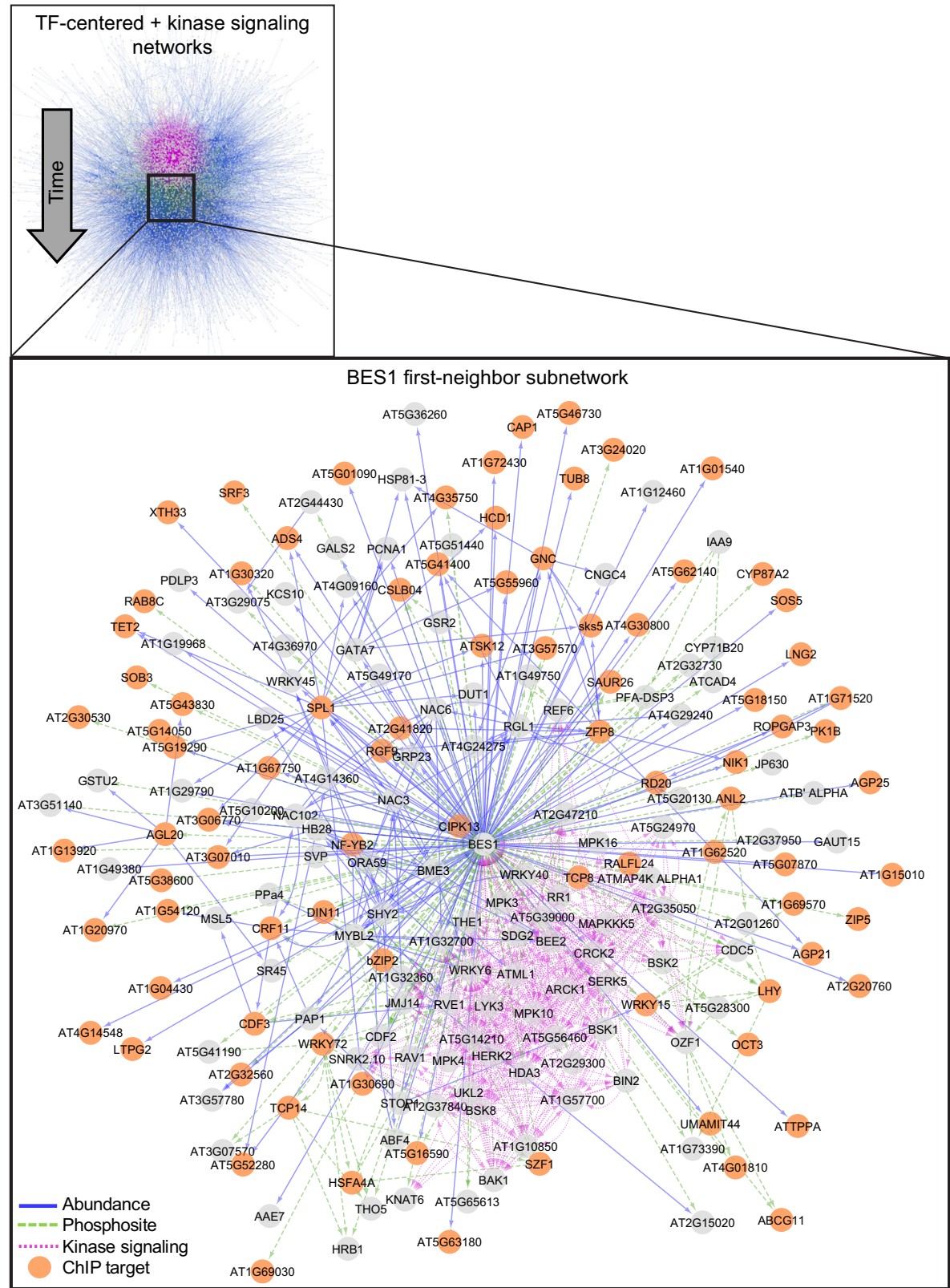

**Fig. 3 Integration of TF-centered and kinase-signaling networks.** (Top) Merged networks. TF abundance, blue solid; TF phosphosite, green dashed; kinase signaling, purple dotted. The network is arranged temporally, with early regulations at the top and later regulations towards the bottom. (Bottom) First-neighbor subnetwork of BES1. Genes are gray circles. Orange genes are BES1 and BZR1 ChIP-Seq targets.

regulation of downstream target genes by TFs including BES1 and BZR1. Our timepoint-specific networks further illustrate which regulations are predicted to occur in early or late BR response (Fig. 2B).

**Multi-omics networks reveal the BIN2-BES1 signaling cascade.** We extracted the first-neighbor network of BES1 to illustrate how our multi-omics profiling and integrative network inference can elucidate the signaling events that occur in response to BR (Fig. 3). This network contains kinases that are predicted to be upstream of BES1 phosphorylation, TFs that are predicted to transcriptionally regulate BES1, and downstream targets of BES1. Our kinase-signaling network predicted that Serine 179 and Serine 180 (S179, S180) on BES1 is downstream of known BR-responsive kinases such as BIN2, BAK1, and BSK1[8,9,44] (Supplementary Figure 3a). The sites are largely conserved in BES1 and its homologs (Supplementary Figure 3b). In order to test if they are BIN2 kinase phosphorylation sites, we mutated the two sites from serine to alanine (phosphorylation null, designated as BES1[S2A]) and tested phosphorylation by BIN2 in an in the vitro kinase assay. While both BES1 and BES1[S2A] were phosphorylated by BIN2, BES1[S2A] was phosphorylated to a lesser extent (Supplementary Figure 3e), suggesting that S179 and S180 are BIN2 phosphorylation sites. We also transiently expressed *BES1-FLAG* and *BES1[S2A]-FLAG* in *Nicotiana benthamiana*. Although a significant amount of *BES1-FLAG* is phosphorylated 1 day after transfection, *BES1[S2A]-FLAG* exists mainly in dephosphorylated form. In addition, when co-expressed with BIN2, the phosphorylated form of wild-type *BES1-FLAG* was increased, whereas the phosphorylated form of *BES1[S2A]-FLAG* was increased to a lesser extent, which further supports the conclusion that S179 and S180 are BIN2 phosphorylation sites (Supplementary Figure 3c).

In order to elucidate the functional consequences of the phosphorylation on S179, S180, we generated transgenic *Arabidopsis* plants overexpressing *BES1-FLAG* or *BES1[S2A]-FLAG*. The majority (20/26: 77%) of the *BES1[S2A]-FLAG* T1 plants have longer leaf petiole and curly leaves, a characteristic of gain-of-function mutants in BR signaling[12]. Conversely, 13 of 14 (93%) *BES1-FLAG* plants grown at the same time did not exhibit these phenotypes (Supplementary Figure 3d). Taken together, our results support our kinase-signaling network prediction that S179 and S180 are BIN2 phosphorylation sites whose phosphorylation negatively impacts BES1 function.

Downstream of kinase signaling, BES1 is predicted to regulate different downstream targets depending on whether its phosphosite levels (green, dashed) or transcript abundance (blue, solid) is used in the SC-ION pipeline. We mined Chromatin Immuno-Precipitation (ChIP)-chip and ChIP-Seq data sets on BES1 and its homolog, BZR1[13–15], and found that 88 out of 144 predicted first-neighbor targets of BES1, in our network (61%), are directly bound by BES1 or its homolog BZR1 (Fig. 3, orange circles; Supplementary Table 1). Some of these validated genes with known roles in BR response include IAA3/SHORT HYPOCO-TYL 2 (SHY2)[45], XYLOGLUCAN:XYLOGLUCOSYL TRANS-FERASE 33 (XTH33)[15,46] and SMALL AUXIN UP RNA 26 (SAUR26)[13]. Thus, we were able to validate the accuracy of our integrative omics network using previously published ChIP data and identify previously unknown BIN2 phosphorylation sites on BES1.

**Network motif analysis predicts TFs involved in BR response.** We next leveraged the network prediction to identify candidate genes involved in mediating the response to BR. We used the Network Motif Score (NMS)[35,36] to classify genes in the TF-centered networks based on their presence in certain biological

motifs, such as feedback and feed-forward loops. Genes with higher NMS have been shown to have a more important role in the biological process of interest[35,36,47–49]. Accordingly, BES1 had the 25th highest NMS score (top 5%), illustrating that we could use the NMS to identify BR-response regulators. We chose three TFs, ANTHOCYANLESS 2 (ANL2), TCX2, and BRONTO-SAURUS (BRON; AT1G75710), that had high (all in the top 35%) NMS in either the TF abundance or phosphosite GRNs (Supplementary Data 5). We then examined subnetworks for each of these TFs, starting with kinase signaling and ending with transcriptional regulation.

SC-ION predicts that our first TF of interest, TCX2, and BES1/BZR1 HOMOLOG 2 (BEH2) regulate each other in a feedback loop (Supplementary Figure 4). In addition, TCX2 has a documented root stem cell division phenotype[35]. Thus, we reasoned that TCX2 may regulate cell division in response to BR and treated the *tcx2-2* and *tcx2-3* mutants with 100 nM BL. In WT plants, the addition of BL causes a dramatic reduction in the root length. However, we did not find a significant difference in BL response in either of the *tcx2* mutant alleles compared to WT (Supplementary Figure 4, Supplementary Table 2).

We then focused on the subnetwork for ANL2, which predicts that ANL2 and BES1 regulate each other in a feedback loop (Supplementary Figure 4). To test if ANL2 may be involved in BR response, we treated WT, *anl2-2*, and *anl2-3* plants with 100 nM BL. We found that *anl2* mutant roots shorten more than WT when treated with BL, suggesting that the *anl2* mutants are hypersensitive to BL (Supplementary Figure 4, Supplementary Table 2). It has also been shown that BL can induce excessive QC divisions in the root[22–24]. Thus, we checked the *anl2* mutant roots for QC divisions as a secondary BR-response phenotype: however, we found that *anl2* mutant roots do not display excessive QC divisions.

**BRONTOSAURUS regulates cell division in response to BR.** Our last TF of interest, BRON, is predicted to be regulated by TFs whose phosphorylation is dependent on multiple BR-signaling kinases such as BAK1, BSKs, and SERKs in our network (Fig. 4a). We obtained a weak (*bron-1*) and a strong (*bron-2*) allele for BRON (Supplementary Figure 5) and examined their BR response as well as their root development. We found that roots from both alleles shorten more in response to BL than the wild type, demonstrating that *bron* mutant roots are hypersensitive to BL treatment (Fig. 4b, Supplementary Table 2). In addition, we observed that *bron* mutants have longer roots with more mer-istematic cells (Supplementary Figure 5), as well as significantly more divisions in the QC (Fig. 4c–e). *bron-2* also displays excessive columella divisions and a higher number of undivided (i.e., actively dividing) cortex endodermis initials (CEI), whereas *bron-1* only shows excessive endodermis divisions, potentially due to its weaker effect on BRON expression (Fig. 4e). This led us to hypothesize that BRON could regulate cell division in response to BR.

To determine the transcripts modulated by BRON, we performed RNAseq on root tissue from the *bron-2* mutant (Supplementary Data 7). We found that a range of BR-responsive genes identified from our time course profiling was enriched in this mutant, particularly those genes repressed by BL after 15 min and induced by BL after 1 or 2 h in our time course ($p < 0.001$) (Supplementary Data 8). Further, we found that most of the genes have lower expression in the *bron-2* mutant, suggesting that BRON transcriptionally activates these genes (Supplementary Figure 5, Supplementary Data 8).

Given the role of cyclins in regulating cell division[50], we specifically examined the expression of cyclins in the *bron-2*

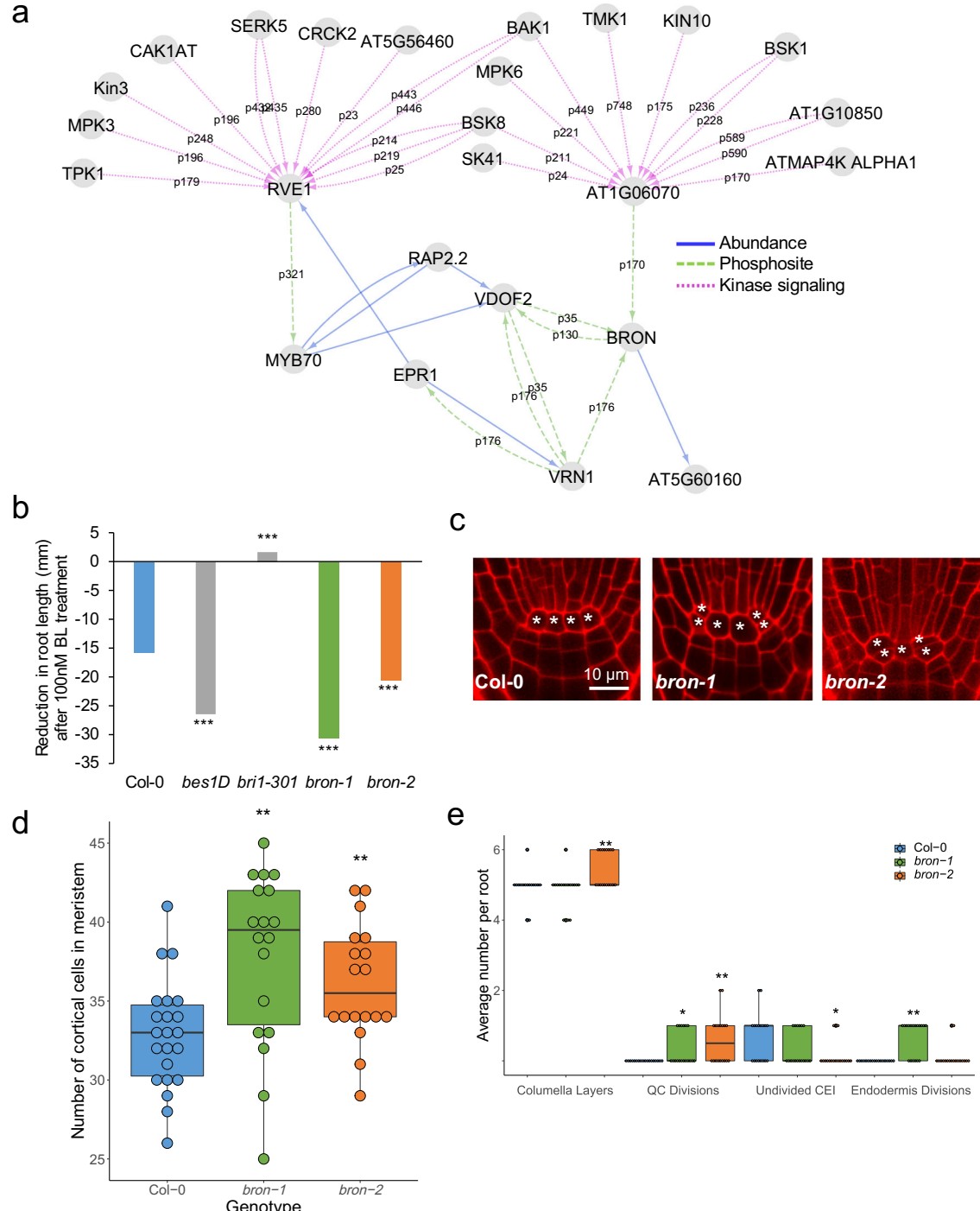

**Fig. 4 *bron* mutant roots are hypersensitive to BL treatment and display excessive cell divisions. a** BRON subnetwork. Edge labels represent the phosphosite used to predict that regulation. **b** *bron* mutants are hypersensitive to 100 nM BL treatment ($n > 27$). **c** Representative images of Col-0 wild type, *bron-1*, and *bron-2* root stem cell niches. * denote Quiescent Center (QC) cells. **d** The number of meristematic cells in 5-day-old roots ($n = 18$). **e** Average number of columella cell layers, QC divisions, undivided Cortex Endodermis Initials (CEI), and endodermis division in 5-day old roots ($n = 18$). **d, e** the centerline represents median; box bounds represent 25th and 75th percentiles; whiskers represent minimum and maximum. Colored dots represent individual data points. **b, d, e** * denotes $p < 0.05$, **$p < 0.01$, ***$p < 0.001$ using a generalized linear model (**b**) or two-tailed Wilcoxon test (**d, e**). For all statistical tests, no multiple testing correction was performed.

mutant. We found that four cyclins are DE in the *bron-2* mutant: three are repressed by BRON (*CYCD3;1, CYCP4;1, CYCP4;2*), whereas one is induced by BRON (*CYCP3;2*). Further, two of these cyclins, *CYCD3;1*, and *CYCP4;1*, are significantly induced by BL at 4 and 8 h after treatment Supplementary Figure 6). Importantly, it has been shown that *CYCD3;1* is induced by BL

and contributes to BL-regulated cell division[21,22]. We also examined the cell-type-specific expression of *CYCD3;1, CYCP4;1*, and *BRON* in the root stem cells and mature root cells[35,51], and we found key differences in where *CYCD3;1* and *CYCP4;1* are co-expressed with BRON. For example, *CYCP4;1* is co-expressed with *BRON* only in the QC. In contrast, *BRON* and *CYCD3;1* are

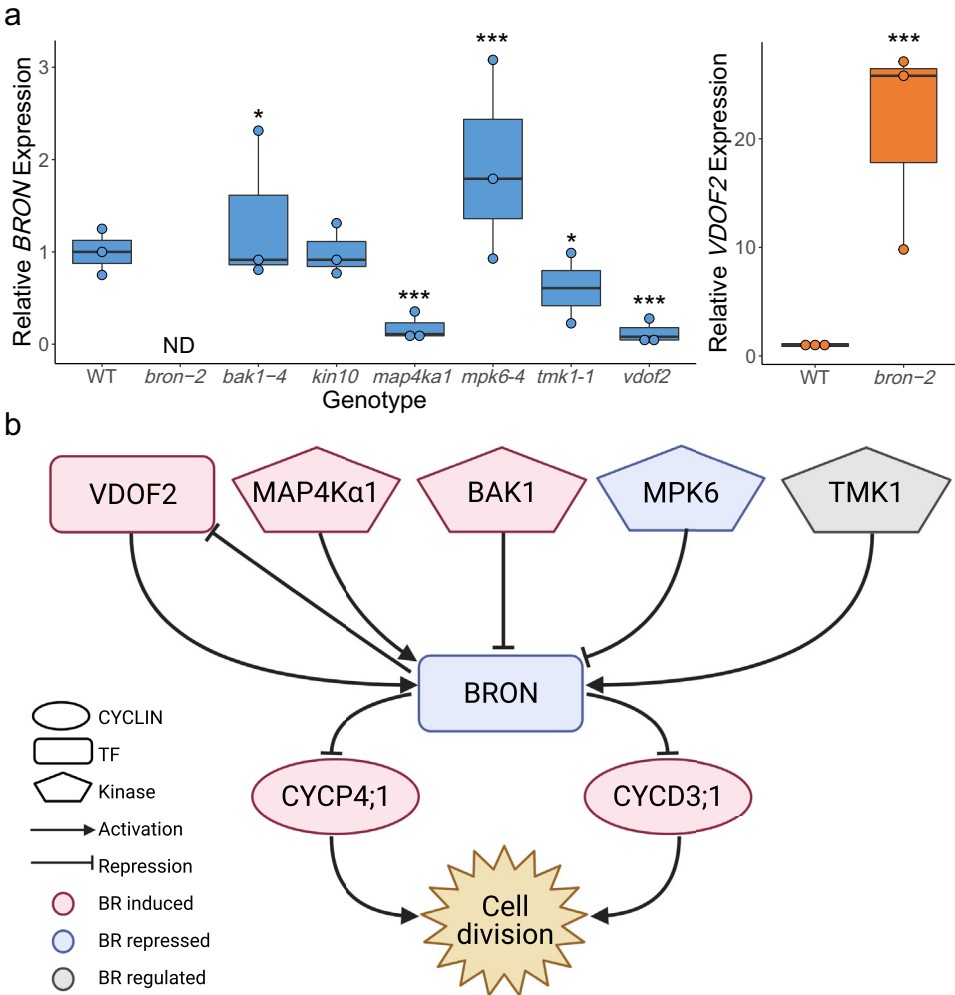

**Fig. 5 BRON regulates cell division in response to BR. a** RT-qPCR of *BRON* (left, blue) and *VDOF2* (right, orange) in mutant lines. Reported values are $2^{-\Delta\Delta CT}$. Centerline represents the median; box bounds represent 25th and 75th percentiles; whiskers represent minimum and maximum. Colored dots represent the average of two technical replicates for each biological replicate ($n = 3$ except for *tmk1-1*, where $n = 3$ were assayed but *BRON* was detected in only $n = 2$). ND: not detected. * denotes $p < 0.05$, *** $p < 0.0001$ using two-tailed *z* test compared with mean and standard deviation of WT. Multiple testing correction was not performed. **b** Proposed model for the role of BRON in BR response. The BR-responsive kinases MAP4Kα1, BAK1, MPK6, and TMK1 regulate *BRON*. In addition, BRON and VDOF2 regulate each other in a negative feedback loop. The coordination of these upstream regulators results in the repression of *BRON* after BL treatment. This repression of *BRON* lifts its repression on the cyclins *CYCD3;1* and *CYCP4;1*, leading to BR-induced cell division.

co-expressed in the Epidermis/Lateral Root Cap (Epi/LRC) initials and the mature columella (Supplementary Figure 6). These results suggest that BR induces the expression of *CYCD3;1* and *CYCP4;1*, and therefore cell division, through the repression of *BRON*.

Our GRN predicts several kinases and TFs as putative upstream regulators of BRON (Fig. 4a). These putative regulators are also responsive to BL in our time course (Supplementary Data 1–3), suggesting that they may be a part of the BR-BRON-CYCLIN signaling cascade. Thus, we measured *BRON* expression via RT-qPCR in mutant lines for six of these putative regulators: BRI1-ASSOCIATED RECEPTOR KINASE (BAK1), SNF1 KINASE HOMOLOG 10 (KIN10), ATMAP4K ALPHA1 (MAP4Kα1), MAP KINASE 6 (MPK6), TRANSMEMBRANE KINASE 1 (TMK1), and VASCULAR-RELATED DOF2 (VDOF2). We found that *BRON* is significantly induced in *bak1-4* and *mpk6-4* mutants and significantly repressed in *map4ka1*, *tmk1-1*, and *vdof2* mutants. These results suggest that BAK1 and MPK6 repress *BRON*, whereas MAP4Kα1, TMK1, and VDOF2 activate *BRON* (Fig. 5a, Supplementary Table 3).

Another prediction in our GRN is the feedback loop between BRON and VDOF2 (Fig. 4a). Feedback loops are one of the developmentally important motifs that we used to calculate the NMS, which is how we chose BRON as a putative regulator of BR response. Accordingly, we also measured *VDOF2* expression via RT-qPCR in *bron-1* and *bron-2* mutants and found that both mutants have significantly higher levels of *VDOF2* (Fig. 5a, Supplementary Figure 5, Supplementary Table 3). This suggests that BRON represses *VDOF2*, forming a negative feedback loop between these two components of the signaling pathway.

Taken together, these results validated our GRN-predicted upstream regulators and informed us about downstream targets of BRON (Fig. 5b). In our network, *BRON* is coordinately regulated by the BR-responsive kinases BAK1, MAP4Kα1, MPK6, and TMK1 as well as the BR-induced TF VDOF2. In addition, BRON represses *VDOF2*, forming a negative feedback loop. The net effect of this regulation is the BR-modulated repression of *BRON*. This lifts the repression on *CYCD3;1* and *CYCP4;1* by BRON, allowing for BR-induced cell divisions in the *Arabidopsis* root.

## Discussion

Here, we used a systems biology approach to unravel the temporal response to BR in *Arabidopsis*. By generating and integrating omics data sets, we were able to quantify how transcript, protein, and phosphorylation levels change in response to BR over time. We found that most of the (phospho)protein response occurs in the earlier timepoints (before 1 h), whereas there are sets of early- and late-responsive transcripts (Fig. 1). This suggested that BR first triggers the phosphorylation of TFs, which then go on to regulate transcripts at later timepoints. Our predictive network reconstructed using SC-ION corroborated this hypothesis, illustrating how BR-responsive kinase signaling leads to phosphorylation of TFs and downstream transcriptional regulation (Fig. 3). Further, our timepoint-specific networks allow us to predict which regulations occur early and late in BR response (Fig. 2).

One well-known example of BR response involves the TFs BES1 and BZR1, which are dephosphorylated in response to BR, allowing them to move to the nucleus and transcriptionally regulate downstream genes. It is well-established that the GSK3 kinase BIN2 phosphorylates BES1 and BZR1 negatively regulate their functions through the promotion of cytoplasmic retention, inhibition of DNA binding, and protein degradation of BES1 and BZR1[1]. Although there are >20 predicted sites, the exact BIN2 phosphorylation sites on BES1 are not functionally characterized. Through our omics profiling and network analysis, we predicted that S179 and S180 in BES1 are potentially phosphorylated by BIN2. Further mutational analysis indicated that the phosphorylation of the two sites contributed to BES1 phosphorylation by BIN2 and are important for BES1 activity. Our study, therefore, identified two key BIN2 phosphorylation sites that negatively regulate BES1 activity and hence BR signaling (Supplementary Figure 1).

We further used our network to predict novel roles for the TFs ANL2 and BRON in BR response (Figs. 4 and 5, Supplementary Figure 4). We found that mutants of both TFs are hypersensitive to BL treatment, but only *bron* mutants display excessive cell divisions. Given *anl2*'s hypersensitivity to BL, it would be interesting to investigate its role in BR response in the future, given that it likely has a different role than BRON. We additionally found a significant overlap between genes DE in the *bron-2* mutant and genes induced or repressed by BR. Specifically, we observed a significant overlap in disease resistance, drought response, and stress response genes which were repressed by BL 15 min after treatment and repressed in the *bron* mutant (Supplementary Figure 5, Supplementary Table 4, Supplementary Data 9). Crosstalk between BR and drought response is well-known and involves other TFs such as WRKY46/54/70, RD26, and TINY[33,52–54], but the temporal aspects of this signaling have not been previously examined. Our observation that these genes are regulated by BLs as early as 15 min after treatment suggests that inhibition of stress responses quickly follows activation of the BR pathway.

We found that cyclin genes, specifically *CYCD3;1* and *CYCP4;1*, are induced by BL and repressed by BRON, suggesting that BR induces these cyclins through its repression of BRON (Fig. 5c). These results are supported by multiple studies which elucidate the role of CYCD3;1 in BR-induced cell division[21,22] and describe how BR specifically induces QC division in the root[22–24]. Although *CYCP4;1* has not been directly implicated in BR response, its co-expression with *CYCD3;1* in both the *bron* mutant and BL time course suggests that it may have a similar function as CYCD3;1 in the *Arabidopsis* root. Further, by mining cell-type-specific transcriptomic data sets[35,51], we gained insight into the cell types in which these cyclins are active. We found that *CYCP4;1* and *BRON* are specifically co-expressed in the QC,

suggesting that BRON may repress QC division through CYCP4;1. In contrast, *CYCD3;1* and *BRON* are co-expressed in the Epi/LRC stem cell initials as well as the mature columella. We found that *bron-2* mutants have excessive columella divisions (Fig. 4e), suggesting that perhaps BRON represses cell division in the columella and lateral root cap through CYCD3;1. It would be of interest in the future to unravel BRON's cell-type-specific repression of cell division in response to BR.

In addition to identifying putative downstream targets of BRON, we also validated our GRN prediction that BAK1, MAP4Kα1, MPK6, TMK1, and VDOF2 regulate *BRON* (Fig. 5). Our network predicts that the kinases in this list may phosphorylate the TF bZIP69 (AT1G06070), which putatively regulates *BRON*. Although we conclude that these kinases do regulate *BRON* expression, it is unclear whether this regulation is direct or indirect. In addition, we showed that all of these regulators are BR-responsive in our time course data set. Some of these regulators already have documented roles in BR response, such as BAK1, which is a key component of the early BR response pathway[7,8]. It has been shown that BES1 is a direct substrate of MPK6[55], and recent results suggest that MPK6 has an important role in BR-autophagy crosstalk[56]. It has also been shown that VDOF2 overexpression lines have higher levels of BR-related transcripts and enrichment of BR-related GO terms[57]. An interesting component of this regulatory network is the negative feedback loop between BRON and VDOF2 (Fig. 5b). *VDOF2* is induced by BL in our time course, and our RT-qPCR results suggest that VDOF2 activates *BRON*. However, our time course also shows that BL represses *BRON*. Based on our results, the repression of *VDOF2* by BRON likely overcomes its activation, resulting in net repression of *BRON* by BL. Future work could investigate the coordinated regulation of *BRON* by this suite of upstream regulators and the resulting effect on BR-induced cell division.

Taken together, our temporal, integrative omics data set, kinase signaling, and TF-centered networks can be used as a resource to identify additional genes like BRON, which are implicated in BR response in *Arabidopsis*.

## Methods

**Plant materials and growth conditions**. The *Arabidopsis* accession Columbia-0 was used as the wild-type control in all experiments. The T-DNA insertion mutants *tcx2-2* (SAIL_808_H08), *tcx2-3* (SALK_021952), *anl2-2* (SALK_000196C), *anl2-3* (SAIL_418_C10), *kin10* (SALK_127939C), *mpk6-4* (SALK_062471C), and *tmk1-1* (SALK_016360) were described previously[35,58–60]. The *bes1-D* and *bri1-301* mutants were also described previously[4,12,23,61]. The T-DNA insertion mutants *at1g75710/bron-1* (SALK_048268), *at1g75710/bron-2* (SALK_046220C), *bak1-4* (SALK_116202C), *map4ka1* (SALK_033601C), and *vdof2* (SALK_130584) were obtained from the Arabidopsis Biological Resource Center (ABRC: https://abrc.osu.edu/).

For the BL time-series profiling experiments, WT seeds were sterilized with 70% EtOH + 0.1% Triton for 15 min, washed with 100% EtOH three times, and plated on 1/2 LS plates with 1% sucrose and 1000 nM BRZ with nylon mesh overlaid on top of the agar. After 3 days of stratification at 4 °C, the plates were placed under continuous light at 22 °C for 7 days. BL treatments were performed by transferring the seedlings on the nylon mesh to 1/2 LS liquid medium with either 1 μM BL or dimethyl sulfoxide (DMSO) for 15 min, 30 min, 1 h, 2 h, 4 h, or 8 h. Four biological replicates were collected for BL- and mock-treated samples at each time point (48 samples total). Samples were blotted dry with Kimwipes, flash-frozen in liquid nitrogen, and ground for 15 min under liquid nitrogen using a mortar and pestle.

For the *bron-2* RNAseq experiment, seeds were wet sterilized using 50% bleach, 10% Tween, and water and stratified at 4 °C for 2 days. Seeds were plated on 1× MS, 1% sucrose plates with Nitex mesh, and grown under long-day conditions (16 h light/8 h dark) at 22 °C for 5 days. Three biological replicates of 10 plates each were collected in RLT buffer, and RNA extraction was immediately performed.

For RT-qPCR experiments, seeds were wet sterilized using 20% bleach and stratified at 4 °C for 2 days. Seeds were plated on nylon mesh on 1/2× LS, 1% sucrose plates supplemented with 1000 nM BRZ, and grown under long-day conditions at 22 °C for 7 days. The seedlings on the mesh were then transferred to agar plates containing 1/2× LS, 1% sucrose for 4 h to simulate mock conditions.

Three biological replicates of 2 plates each were collected and flash-frozen in liquid nitrogen prior to RNA extraction.

**Antibodies**. The anti-BES1 antibody was generated in ref. [14] and used at a dilution of 1:5000. The anti-FLAG antibody was obtained from Sigma-Aldrich (Cat #F7425, RRID: AB_439687) and used at a dilution of 1:1500. The anti-GFP antibody was generated in ref. [62] and used at a dilution of 1:1000.

**BES1 western blot**. To monitor BES1 protein levels and phosphorylation status, BL treatment was performed as described above. Approximately 100 mg of ground tissue powder was resuspended directly in 300 μL 2× SDS sample buffer (100 mM Tris-Cl, pH 6.8, 4% (w/v) sodium dodecyl sulfate, 0.2% (w/v) bromophenol blue, 20% (v/v) glycerol and 200 mM dithiothreitol) before sodium dodecyl sulphate–polyacrylamide gel electrophoresis (SDS-PAGE) and western blotting using anti-BES1 antibody at a dilution of 1:5000[14].

**RNA sequencing and data analysis**. For the BR timecourse, total RNA was extracted using Zymo Direct-zol kit (Zymo Research). RNA concentration was measured with Qubit RNA HS assays (ThermoFisher #Q10213) and integrity checked with AATI Fragment Analyzer with Standard Sensitivity RNA Analysis Kit (DNF-489-0500). Approximately 500 ng of RNA was used for library construction via the QuantSeq 3′ mRNA-Seq Library Prep FWD Kit for Illumina. Sequencing was performed on a HiSeq 3000 with 50 bp single-end reads. Raw sequencing data are deposited at the Gene Expression Omnibus (https://www.ncbi.nlm.nih.gov/geo/query/acc.cgi?acc=GSE147589). Reads were mapped to the TAIR10 genome using the STAR aligner[63]. Differential expression between BL-treated and mock seedlings at each time point was performed using PoissonSeq[64] using a *q* value cutoff of 0.05 and a fold-change cutoff of 1.25.

For *bron-2* transcriptional profiling, total RNA was isolated from ~2 mm of 5-day-old Col-0 and *bron-2* root tips using the RNeasy Micro Kit (Qiagen). cDNA synthesis and amplification were performed using the NEBNext Ultra II RNA Library Prep Kit for Illumina. Libraries were sequenced on an Illumina HiSeq 2500 with 100 bp single-end reads. Reads were mapped to the TAIR10 genome using Cufflinks[65]. Differential expression was performed using PoissonSeq with a *p* value cutoff of 0.05. Raw sequencing data are deposited at the Gene Expression Omnibus (https://www.ncbi.nlm.nih.gov/geo/query/acc.cgi?acc=GSE157000).

**Protein extraction and digestion**. The proteomics experiments were carried out based on established methods[28,29,31]. Protein was extracted from aliquots of the tissue used for transcriptome profiling and digested into peptides with trypsin and LysC using the phenol-FASP method detailed in[28,31]. The resulting peptides were desalted using 50 mg Sep-Pak C18 cartridges (Waters), dried using a vacuum centrifuge (Thermo), and resuspended in 0.1% formic acid. Peptide amount was quantified using the Pierce BCA Protein assay kit.

**TMT labeling**. The TMT-labeling strategy used in this experiment is provided in Supplementary Table 5. In all, 45 μg of peptides were taken from each individual sample, pooled, and then split into two pooled references. TMT10plex™ label reagents (ThermoFisher, Lot #UD280154) were used to label 200 μg of peptides, from each sample or pooled reference, at a TMT:peptide ratio of 0.2:1[31]. After 2 h incubation at room temperature, the labeling reaction was quenched with hydroxylamine. Next, the ten samples were mixed together, and an aliquot of 75 μg of peptides was reserved for protein abundance profiling, and the remaining peptides were used for phosphopeptide enrichment and stored at −80 °C. Labeling efficiency was checked by performing a 60-minute 1D run on 200 ng of TMT-labeled peptides. All samples had labeling efficiencies ≥94.7% (Supplementary Table 5).

**Phosphopeptide enrichment**. The TMT-labeled phosphopeptides were first enriched using the High-Select TiO₂ Phosphopeptide Enrichment Kit (Thermo) using the manufacturer's protocol. The High-Select Fe-NTA Phosphopeptide Enrichment Kit (Thermo) was then used on the flowthrough from the TiO₂ enrichment to enrich additional phosphopeptides. The manufacturer's protocol for the Fe-NTA kit was used except the final eluate was resuspended with 50 μL 0.1% formic acid. The eluates from the TiO₂ and Fe-NTA enrichments were combined and stored at −80 °C until analysis by LC-MS/MS.

**LC-MS/MS**. An Agilent 1260 quaternary HPLC was used to deliver a flow rate of ~600 nL min⁻¹ via a splitter. All columns were packed in-house using a Next Advance pressure cell, and the nanospray tips were fabricated using a fused silica capillary that was pulled to a sharp tip using a laser puller (Sutter P-2000). In all, 25 μg of TMT-labeled peptides (non-modified proteome), or ~10 μg TiO₂-enriched peptides (phosphoproteome), were loaded onto 20 cm capillary columns packed with 5 μM Zorbax SB-C18 (Agilent), which was connected using a zero dead volume 1 μm filter (Upchurch, M548) to a 5 cm long strong cation exchange (SCX) column packed with 5 μm PolySulfoethyl. The SCX column was then connected to a 20 cm nanospray tip packed with 2.5 μM C18 (Waters). The three sections were joined and mounted on a Nanospray Flex ion source (Thermo) for online nested peptide elution. A new set of columns was used for every sample. Peptides were

eluted from the loading column onto the SCX column using a 0 to 80% acetonitrile gradient over 60 min. Peptides were then fractionated from the SCX column using a series of 18 and 6 salt steps (ammonium acetate) for the non-modified proteome and phosphoproteome analysis, respectively. For these analyses, buffers A (99.9% H₂O, 0.1% formic acid), B (99.9% ACN, 0.1% formic acid), C (100 mM ammonium acetate, 2% formic acid), and D (2 M ammonium acetate, 2% formic acid) were utilized. For each salt step, a 150-minute gradient program comprised of a 0–5 min increase to the specified ammonium acetate concentration, 5–10 min hold, 10–14 min at 100% buffer A, 15–120 min 10–35% buffer B, 120–140 min 35–80% buffer B, 140–145 min 80% buffer B, and 145–150 min buffer A was employed.

Eluted peptides were analyzed using a Thermo Scientific Q-Exactive Plus high-resolution quadrupole Orbitrap mass spectrometer, which was directly coupled to the high-performance liquid chromatography (HPLC). The data-dependent acquisition was obtained using Xcalibur 4.0 software in positive ion mode with a spray voltage of 2.10 kV and a capillary temperature of 275 °C and an RF of 60. MS1 spectra were measured at a resolution of 70,000, an automatic gain control (AGC) of 3e6 with a maximum ion time of 100 ms and a mass range of 400–2000 m/z. Up to 15 MS2 were triggered at a resolution of 35,000 with a fixed first mass of 120 m/z for phosphoproteome and 115 m/z for proteome. An AGC of 1e5 with a maximum ion time of 50 ms, an isolation window of 1.3 m/z, and normalized collision energy of 33. Charge exclusion was set to unassigned, 1, 5–8, and >8. MS1 that triggered MS2 scans were dynamically excluded for 45 or 25 s for phospho- and non-modified proteomes, respectively.

**Proteomics data analysis**. The raw data were analyzed using MaxQuant version 1.6.7.0[66]. Spectra were searched, using the Andromeda search engine in MaxQuant[67] against the Tair10 proteome file entitled "TAIR10_pep_20101214" that was downloaded from the TAIR website (https://www.arabidopsis.org/download_files/Proteins/TAIR10_protein_lists/TAIR10_pep_20101214) and was complemented with reverse decoy sequences and common contaminants by MaxQuant. Carbamidomethyl cysteine was set as a fixed modification while methionine oxidation and protein N-terminal acetylation were set as variable modifications. The phosphoproteome "Phospho STY" was also set as a variable modification. The sample type was set to "Reporter Ion MS2" with "10plex TMT selected for both lysine and N-termini". TMT batch-specific correction factors were configured in the MaxQuant modifications tab (TMT Lot UD280154). Digestion parameters were set to "specific" and "Trypsin/P;LysC". Up to two missed cleavages were allowed. A false discovery rate, calculated in MaxQuant using a target-decoy strategy[68], less than 0.01 at both the peptide spectral match and protein identification level was required. The "second peptide" option to identify co-fragmented peptides was not used. The match between runs feature of MaxQuant was not utilized. Raw proteomics data have been deposited on MassIVEand can be accessed at the link ftp://massive.ucsd.edu/MSV000085606/

Statistical analysis was performed using TMT-NEAT Analysis Pipeline version 1.4 (https://doi.org/10.5281/zenodo.5237316). This pipeline takes the "proteinGroups" (protein abundance) or "Phospho(STY)Sites" (phosphoproteome) tables output from MaxQuant as well as a metadata file detailing the TMT-labeling scheme and sample information as input. Example input files are provided in the GitHub repository. First, the MaxQuant output table is trimmed to only include the needed information for statistical analysis, and the columns are re-labeled using the provided sample information. Contaminants are removed at this stage. Next, data are normalized using the sample loading normalization and internal reference normalization methods such that samples can be compared across runs[32]. Quantitative plots such as boxplots, hierarchical clustering, and principal components analysis are provided for quality control. Finally, statistical analysis is performed using PoissonSeq[64], and histograms of *p*- or *q* value distributions are generated. Proteins and phosphosites were categorized as differentially accumulating between the mock and BL-treated samples at each time point if they had a *p* value <0.05 and fold-change >1.1.

**GO analyses**. GO analysis on the DE transcripts, protein, and phosphoproteins were performed using PANTHER[69]. Genes were separated depending on whether they were induced or repressed by BL in early (1 h or prior) or late (after 1 h) time points. Biological process GO terms were considered significantly enriched if they had a corrected *p* value ≤0.05 (Supplementary Data 4).

**GRN inference and validation**. TF-centered GRNs were inferred using SC-ION version 2.1 (https://doi.org/10.5281/zenodo.5237310). SC-ION builds on the RTP-STAR pipeline[35,36] by incorporating DTW and ICA clustering[38,39] and integration of different data types. SC-ION uses an adapted version of GENIE3[37], which allows for the separate regulator and target data matrices[34]. This allows the user, for example, to use protein abundance data for regulators (TFs) and transcript data for targets (all genes). SC-ION takes regulator and target lists and regulator and target data matrices as input. In addition, SC-ION takes a clustering matrix, which can be different from the regulator and target data matrices. This allows the user to cluster genes based on different data types. A version of SC-ION without this clustering step is also available. SC-ION outputs a table of the predicted regulations as well as a weight for each edge, where a higher weight indicates higher confidence in that inferred edge[37]. This table can be imported into software such as

Cytoscape[70] for network visualization. Test input files for SC-ION are provided on GitHub.

Two TF-centered networks were inferred (Supplementary Figure 7). In the first network, TF protein abundance (when quantified) or TF transcript abundance (when cognate protein was not quantified) was used as the "regulator" value to infer their "target" genes' transcript abundance. In the second phosphosite network, we inferred the transcript levels of "target" genes using TF phosphosite intensities as "regulators." In both the networks, only the regulators and targets DE at each time point were used in these individual subnetworks. Next, the clustering matrix was constructed by combining the "regulator" (TF abundance) and "target" (transcript levels) data so that genes were clustered based on the protein/ phosphosite levels of the regulators and transcript levels of the targets. So, for example, TF protein abundance is clustered with the transcript abundance of its potential target genes. Clustering was then performed using the DTW method given the temporal nature of our data. After clustering, six subnetworks were inferred for each TF-centered network, where each subnetwork represents the regulations predicted to happen using only one timepoint (15 min, 30 min, 1 h, 2 h, 4 h, 8 h). This allows us to denote which regulations are predicted to happen early or late in BR response. Further, TFs are only predicted to regulate transcripts in their same cluster. This allows the clustering to inform the network prediction by reducing the possible number of edges. The six subnetworks were combined in a union in Cytoscape to form the final abundance and phosphosite networks. Cytoscape was also used to create the merged abundance, phosphosite, and kinase-signaling networks (Supplementary Data 5).

We have previously used the Normalized Motif Score (NMS) to predict biologically important genes in GRNs from *Arabidopsis*[35,36]. Four different motifs were used to calculate the NMS for the merged abundance and phosphosite networks separately: feed-forward loops, feedback loops, diamond motifs, and bi-fan motifs. First, the number of times a gene appeared in each motif was counted using the NetMatchStar app[71] in Cytoscape. Then, the counts were normalized to a scale from 0 to 1 and summed to calculate the NMS for each gene.

The kinase-signaling regulatory network was inferred using a previously reported correlation-based approach[40]. Kinases with phosphosites in the activation loop, also called p-loop, the domain that was DE in response to BL were used as the potential regulators. All genes with phosphosites that were DE in response to BL were used as the potential targets. We used phosphosite intensity rather than abundance for the regulators as it has been shown that phosphosite intensity has greater predictive power[40] (Supplementary Figure 2). Pearson and Spearman correlations were calculated for each regulator-target pair, and edges were kept for those pairs with Pearson correlation ≥0.5 or Spearman correlation ≥0.6 (Supplementary Figure 7).

Activation loop domains, also called p-loop domains, in protein kinases, were identified using a modified version of the pipeline described in[56] as follows. All 35,386 protein sequences available in the TAIR10 annotation were searched for kinase domains using The National Center for Biotechnology Information batch conserved domain search tool[72]. From this list of 1522 proteins with identified kinase domains, 878 were also annotated with activation loop (p-loop) coordinates by the search tool. The kinase domains of proteins lacking the p-loop coordinates were aligned using MAFFT[73]. The resulting alignment was manually searched for the well-conserved p-loop beginning and end motifs. An extra 482 p-loop coordinates were obtained, for a total of 1360 protein kinases with p-loop coordinates.

Targets in the first-neighbor BES1 subnetwork (Fig. 3) were validated using a list of targets shown to be bound by BES1 or its homolog BZR1 in various ChIP studies[13–15]. The results of this validation are reported in Supplementary Table 1.

Supplementary Figure 2 For more details on the GRN inference methodology, please see Supplementary Methods.

**Mutant BES1 cloning and protein level detection**. The two Serine (S179 and S180) to Alanine mutations were introduced by two-step PCR using the primers provided in Supplementary Table 6. Two BES1 fragments were generated and combined to form the full-length mutant BES1. The full-length mutant BES1 was then cloned to *Pro35S:FLAG* vector to generate *Pro35S:BES1^{S2A}-FLAG*. *Pro35S:BES1-FLAG* was subcloned from *Pro35S:BES1-GFP*[12].

The in vitro kinase assay was based on ref.[74]. Specifically, the 0.2 μg of GST-BIN2 and 2 μg of MBP-BES1 or MBP-BES1^{S2A} were used in 20 μl reactions, and the reactions were stopped by adding an equal volume of 2× SDS buffer at different time points (5 min, 10 min, 20 min, 30 min, 60 min) to observe the phosphorylation dynamics. The in vitro kinase reactions were then resolved on 8% SDS-PAGE incorporated with phostag reagent. The phostag gel was made following the manufacturer's instructions (AAL-107, http://www.Phos-tag.com). SYPRO Ruby protein gel stain was used to stain the gel following the manufacturer's instructions (Invitrogen). The quantification of BES1-P/BES1 ratio and BES1^{S2A}-P/BES1^{S2A} ratio was performed in ImageJ.

For transient expression in *Nicotiana benthamiana*, agrobacterium containing *Pro35S:BES1-FLAG* or *Pro35S:BES1^{S2A}-FLAG* was infiltrated to mature *N. benthamiana* leaves. Agrobacterium containing *Pro35S:YFP-BIN2* was used for co-expression. Leaf discs were collected 24 h after infiltration and flash frozen in liquid nitrogen. The samples were ground in 2× SDS buffer and resolved on SDS-PAGE. Anti-Flag antibody at a dilution of 1:1500 was used for western blotting.

**BR phenotyping methods**. BL phenotyping was carried out as based on[75] as follows. Seeds were sterilized for 4 h in a Nalgene Acrylic Desiccator Cabinet (Fisher Scientific, 08-642-22) by mixing 200 mL bleach (8.25% sodium hypochlorite) with 8 mL concentrated hydrochloric acid to generate chlorine gas. Seeds were then resuspended using 0.1% agarose solution for plating. Control (BL0; DMSO solvent only) or BL100-treated (100 nM Brassinolide; BL, Wako chemicals) were plated on ½ LS plates supplemented with 1% (w/v) sucrose. After seeds were plated, the plates were sealed with breathable tape (3 M Micropore) and placed in the dark at 4 °C for 5 days for stratification. Plants were grown for 7 days at 22 °C under continuous light. Plates were imaged with an Epson Perfection V600 Flatbed Photo scanner at a resolution of 1200 DPI and root length was then measured in ImageJ.

***bron* root phenotyping**. Confocal imaging was performed on a Zeiss LSM 710. Cell walls were counterstained using propidium iodide. The number of meristematic cells and cell divisions were manually counted.

**RT-qPCR**. Total RNA was isolated 1 cm of 7-day-old root tips using the Zymo Direct-zol kit (Zymo Research). RT-qPCR was performed with SYBR green (Invitrogen) using a CFX96 Real-Time PCR System (BioRad) with 40 cycles. Data were analyzed using the $2^{-\Delta\Delta CT}$ ($C_T$: cycle threshold) method and normalized to the expression of the reference gene AT4G34270[76]. RT-qPCR was performed on two technical replicates of three independent RNA samples (biological replicates). $C_T$ values were normalized between runs based on the average $C_T$ values for the reference gene in the WT samples. Primers used for qPCR are provided in Supplementary Table 6. A melt curve was performed all on primer pairs to confirm gene-product specificity.

**Statistics**. A generalized mixed linear model with penalized quasi-likelihood (glmmPQL in R) was used to determine the genotype × treatment interaction $p$ values for the BL phenotyping experiment. In this model, the genotype and treatment effects were considered fixed, and the experiment date and plate number were considered random effects with a Gaussian error distribution. Hypergeometric and/or Chi-squared tests were used for test for enrichment in the RNAseq DE gene lists. For *bron* mutant root phenotyping, a two-tailed Wilcoxon test was used for statistical significance as some of the data did not follow a normal distribution. For RT-qPCR, a $z$ test was used to compare each sample to the mean and standard deviation of the expression of the gene in WT. Hypergeometric testing was performed in R using the phyper function. Chi-squared, Wilcoxon, and $z$ tests were performed using JMP Pro 15 (jmp.com). To select $p$- and $q$ value cutoffs for the large-scale omics experiments, we used the distribution of $p$- and $q$ values generated from the statistical tests based on refs.[77,78].

**Reporting summary**. Further information on research design is available in the Nature Research Reporting Summary linked to this article.

## Data availability

Raw sequencing data are deposited at the Gene Expression Omnibus with accession numbers GSE147589 and GSE15700. Raw proteomics data have been deposited on MassIVE with accession number MSV000085606. Source data are provided with this paper.

## Code availability

Quantitative proteomics statistical analysis was performed using TMT-NEAT Analysis Pipeline version 1.4 (https://doi.org/10.5281/zenodo.5237316)[79]. TF-centered GRNs were inferred using SC-ION version 2.1 (https://doi.org/10.5281/zenodo.5237310)[80].

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

## Acknowledgements

We thank Dr. J. Mitch Elmore for his assistance testing SC-ION and implementing ICA clustering, Dr. Yvon Jaillais for providing the *tmk1-1* mutant seeds, and Dr. Dior Kelley for the inspiration behind the name BRONTOSAURUS. The images presented herein were generated using the instruments and services at the Cellular and Molecular Imaging Facility at North Carolina State University. Next-generation sequencing was performed by the Genomic Sciences Laboratory at North Carolina State University and the Iowa State University DNA Facility. The illustrations in Figs. 2a, 5a, and Supplementary Figure 7 were created using BioRender.com. N.M.C. is supported by a USDA-NIFA Postdoctoral Research Fellowship (2019-67012-29712). T.M.N. is supported by the National Science Foundation Postdoctoral Research Fellowships in Biology Program (Grant no. IOS-2010686). R.S. is supported by an NSF CAREER Grant (MCB-1453130) and the North Carolina Agricultural & Life Sciences Research Foundation at North Carolina State University's College of Agricultural and Life Sciences. The research is supported by grants from NSF (MCB 1818160) and from Plant Sciences Institute from Iowa State University to Y.Y. and J.W.W.

## Author contributions

N.M.C., T.M.N., Y.Y. and J.W.W. conceived and designed experiments. T.M.N. and P.W. performed BES1 western blot, collected plant tissue for integrative omics, and extracted RNA for QuantSeq. N.M.C., G.S. and J.W.W. prepared samples for and performed quantitative proteomics. N.M.C. analyzed QuantSeq and quantitative proteomics data and performed network inference. C.M. annotated activation loop sites for kinases. H.G. performed experiments on BES1 phospho-null mutant. P.W. performed BL root phenotyping and N.M.C. analyzed the data. N.M.C. and R.S. performed root phenotyping on *anl2* and *bron* mutants as well as RNASeq on *bron* mutants. N.M.C. and C.T.V. performed RT-qPCR on mutants. N.M.C. and J.W.W. designed all the main and supplemental figures. N.M.C. and J.W.W. wrote the paper, and all co-authors edited the paper.

## Competing interests

The authors declare no competing interests.
