## [Peer Review File · Nature Communications]

Integrated omics networks reveal the temporal signaling events of brassinosteroid response in ArabidopsisEditorial Note: This manuscript has been previously reviewed at another journal that is not operating a transparent peer review scheme. This document only contains reviewer comments and rebuttal letters for versions considered at *Nature Communications*.

REVIEWER COMMENTS

Reviewer #1 (Remarks to the Author):

The manuscript by Clark et al, reports an integrated omics study that profiled temporal changes in the abundance of transcript/protein and phosphorylation status using wild-type Arabidopsis seedlings that were pretreated with BRZ [a highly-specific inhibitor of brassinosteroid (BR) biosynthesis] followed with treatment with mock solution or 1 micromolar of brassinolide (BL, the most active member of the BR family) for different durations. The authors developed a network inference pipeline, named Spatiotemporal Clustering and Inference of Omics Network (SC-ION), to integrate data of their transcriptomic/proteomic/phosphoproteomic analyses of wild-type Arabidopsis seedlings treated with or without BL, creating a transcriptional factor(TF)-centered network and a kinase-centered network. Merging of the two networks revealed “the temporal signaling events of brassinosteroid response in Arabidopsis” with kinases at early events of the BR signaling followed by changes in abundance/phosphorylation status of TFs and subsequent transcriptional regulation of thousands of Arabidopsis genes. The BR signaling pathway is one of the best studied plant signaling processes. Previous extensive genetic, biochemical, and omics studies have established that BR signaling is executed by a protein phosphorylation-mediated biochemical cascade that regulates the abundance, subcellular locations, and biochemical activities of members of the BES1/BZR1 family. Loss-of-function mutations in BRI1, simultaneous elimination of all 6 BES1/BZR1 members, or gain-of-function mutations in BIN2 (an Arabidopsis homolog of the mammalian GSK3 kinases, which phosphorylates and negatively regulates BES1/BZR1) result in almost identical dwarf phenotypes. Thus, the current study merely confirmed what was already known for the BR signaling pathway for quite some time now.

One of the major concern that I have for this study is the lack of an important negative control, an Arabidopsis triple mutant lacking BRI1 and its two homologs, BRL1 and BRL2, which initiate BR signaling in Arabidopsis with BRI1 playing a predominant role and the other two being involved in vascular development. This control is extremely important to rid of phosphorylation sites (differentially expressed genes or changes in protein abundance) that might be caused by non-specific stress-response pathways (due to exogenous applications of BL at high concentrations). An earlier study (Zhang et al., 2009 PNAS 106:4543) showed that an aqueous buffer without any hormone could stimulate BES1 phosphorylation via an abscisic acid-mediated stress signaling pathway.

The most interesting part of the study is discovery of BRONTOSAURUS (BRON) as a transcription factor that regulates cell division in Arabidopsis roots. Previous studies done in Ana Cano-Delgado’s laboratory

showed that BR controls meristem size and QC division through BES1. While the complete elimination of 6 members of the BES1/BZR1 family [BES1, BZR1, and 4 BES1 homologs (BEHs)] resulted in a morphological phenotype identical to that of a null *bri1* mutant, no study was done to determine the impact of the sextuple *bes1/bzr1/beh* mutation on the root growth. It is possible that additional transcription factors, such as BRON, are involved in BR-regulated root growth via a BES1/BZR1-independent manner. The SC-ION analysis revealed that BRON is a target of At1G06070 (also known as bZIP69, a member of the Arabidopsis bZIP transcription factor family), which is likely regulated by kinases known to be involved in early signaling events, such as BAK1, a coreceptor for BRI1, and two members of the BR signaling kinase (BSK) family involved in BR signaling from BRI1/BAK1 to BIN2. Additional experiments are needed to fully establish the operation of the BR-activated kinase(s)-bZIP69-BRON cascade in BR-stimulated root growth. The presented results on BRON are too preliminary to support the existence of this cascade. Besides, the kinases that connected to bZIP69 are also known to be involved in plant stress response, such as SK41, BSK1, KIN10, and BAK1.

Additional comments:

The authors should double check the numbers that were stated in the abstract. Are they absolutely sure that their transcriptome experiments profiled a total of 32,549 transcripts and their phospho-proteomic assays detected a total of 26,950 phosphorylation sites? There are many genes with “1.39E-17” (meaning no detectable expression??) listed across all cells in Supplementary Table 1, and I think that those genes should be eliminated from their final transcript counts. There are many repetitive phosphopeptides with identical phosphorylated residues in Supplementary Table 3, and a careful analysis of these peptides are needed to get accurate count of detected phosphorylation sites (especially those with statistically significant changes IN RESPONSE to BL treatment, which should be included in a separate sheet of the excel file).

There are at least 16 additional BIN2 substrates (other than the 6 members of the BES1/BZR1 family) (Youn & Kim, 2015, *Mol Plant* 8:552-565), and BR treatment is expected to alter their phosphorylation status. The authors should discuss how many of these previously demonstrated BIN2 substrates were identified in their phosphoproteomic experiments given the large number (26,950) of detected phosphorylation sites. The immunoblot experiments shown in this manuscript and previously published studies suggested the presence of 20-25 putative BIN2 phosphorylation sites on BES1/BZR1. An earlier study (Ryu et al., 2010 *Mol Cells* 29:283) that utilized site-directed mutagenesis showed that the phosphorylation of Ser171 and Thr175 regulates the interaction of BES1 with 14-3-3 proteins for nuclear export while an earlier phosphoproteomic study (Nakagami et al., 2010 *Plant Physiol* 153:1161) detected at least 6 phosphorylation sites in BES1, including Ser169, Ser171, Ser179, Ser180, Thr182, and Ser183. The studies on Ser171 of BES1 and Ser173 of BZR1 revealed the important role of this conserved Ser residue in regulating the subcellular localization of BES1/BZR1, which is consistent with the detected phenotypes shown in Fig. S3E. A functional study that tests the role of Ser179/180 should be performed with a mutant BES1 construct carrying the Ser179Ser180-AA double mutation (instead of the Ser171Ser179Ser180-AAA triple mutation) in order to conclude that “Our study therefore identified

three key BIN2 phosphorylation sites that negatively regulate BES1 activity and hence BR signaling". The immunoblots shown in Fig. S3, C&D are not sufficient to conclude that Ser179/180 are BIN2 phosphorylation sites given that the phosphorylation experiments were performed with the tobacco transient expression system, which contains many kinases and phosphatases. An in vitro phosphorylation assay with purified BIN2 and BES1 (wild-type plus mutated ones) coupled with Phos-tag SDS-PAGE is needed to confirm that the two sites are indeed phosphorylated by BIN2 because the Ser179Ser180 sites do not conform to the S/TxxxS/T consensus phosphorylation motif of known GSK3 kinases.

I am also curious if the phosphoproteomic experiments detected any phosphopeptide of BRI1, which was known to be auto/trans-phosphorylated at >30 sites. BRI1 is a quite abundant protein in Arabidopsis and the reported phospho-proteomic assays did detect phosphorylation sites of many membrane-localized receptor-like kinases.

Reviewer #2 (Remarks to the Author):

The article entitled "Integrated omics networks reveal the temporal signaling events of brassinosteroid response in Arabidopsis" describes a systems biology approach to integrate multi-omic datasets and unravel the molecular signalling events of BR response in Arabidopsis. The BR signalling pathways are very well studied in Arabidopsis, however having a good integrative network analysis applied to multi-omics data in a signalling pathway is new. The authors combined in this paper not only large scale bioinformatics analysis but also showed functional data for a few newly identified genes in the BR signalling. The paper is well written and clear.

I have minor comments to improve the content of the paper.

- What I missed in the paper is the transactivation (early signaling cascade in the BR signaling) of the kinase domains of the receptors involved in the BR perception for example BRI1, BAK1 and others.

- I would like to see more explanation on the Network patterns observed and on conserved network regulations (can you explain using these feedforward and feedback loops how the application of BL will act on the different TFs in the BL transcription network to regulate the signalling response? You have the data so use these networks to describe the interaction and signaling (not just provide a figure with multiple interactions but discuss the interactions and the patterns you see). For example can you make predictions based on the network motifs you observed about how the signaling will change if one TF (for example BRON is not functional)? What will happen to the network observed, can you give an example in combination with your functional data for BRON (including RNA-seq of the mutant).

Reviewer #3 (Remarks to the Author):

This manuscript describes RNAseq and TMT labelled phosphoproteomics revealed the molecular signaling events of brassinosteroids(BR) response in Arabidopsis. In this study, total 32,549 transcripts, 9,035 protein groups, and 26,950 phosphorylation sites from Arabidopsis seedlings treated with brassinolide for six different lengths of time were integrated by a network inference pipeline called Spatiotemporal Clustering and Inference of Omics Networks (SC-ION). Phosphorylation sites on BES1 were experimentally validated their importance. For further study, functional studies of BRONTOSAURUS (BRON) as a transcription factor that regulates cell division were performed. Overall, proteomics studies were well performed and find interesting clues for phosphorylation site of BES in BR signaling.

1. Authors used 10plex TMT kit for labelling of tryptic digested peptides, but Arabidopsis seedlings treated with brassinolide for six different lengths of time were used for proteomics study. Authors should provide more detailed TMT labelling condition and need pooling sets for normalization of quantification.
2. Tryptic digestion and TMT labelling efficiency are required.
3. LC-MSMS but not LC/MS-MS (line 445)
4. I suggest that there are a weak correlation between proteomic studied and functional studies of BRONTOSAURUS (BRON).

Reviewer #4 (Remarks to the Author):

This paper outlines the generation of time series data on Arabidopsis transcript, protein and peptide phosphorylation in response to brassinosteroid treatment of seedlings. This data is then used in network inference to generate networks predicting the impact of transcription factor abundance (protein or transcript) on transcript levels of downstream target genes, or to predict the impact of peptide phosphorylation at specific sites on downstream transcript abundance. From this network the authors identify two phosphorylation sites in BES1 phosphorylated by BIN2; the role of ANL2 in root growth inhibition by BR; a similar role for BRON with additional impacts on root cell division; and profile gene expression in bron mutant roots compared to wildtype. The incorporation of time series peptide phosphorylation data with transcript/protein abundance is an interesting approach and provides plenty

of predictions for experimental testing. However, unfortunately the prediction of the phosphorylation of BES1 by BIN2 could have been made from the phosphorylation data alone, given the known role of BIN2 in phosphorylating another site in BES1 already. The choice of transcription factors for experimental testing is not explained and the findings on the role of ANL2 are minimal. BRON is investigated in slightly more detail but the gene expression profiling does not yield obvious potential mechanisms (especially given the lack of any cell cycle gene enrichment). The manuscript is well written in places but considerable details are missing, and key findings (e.g. around timing of BR response) are not at all clear from the current presentation and explanation. In summary, the data sets are valuable and will no doubt be of value to the community but evidence for the relevance and importance of the network inference is not strong, and the current manuscript does not provide significant advance in our understanding of BR signaling and response. I have provided further information below.

Major points:

1. I could not see any information on the normalisation process for the protein data. Are internal references used? Whole proteome methods?
2. The identification of differentially expressed genes is not clear. Do the authors use the time series nature of the data set? Or simply do pairwise comparisons at each time point? I think it is the latter in which case it would be useful to utilise the time series design to refine these.
3. The list of differentially expressed genes is not particularly rigorous. It would be useful to have information on how many DEGs were specific to a single time point – and greater confidence could be placed in DEGs that were DE over multiple time points. In addition, no fold change threshold has been used. Out of 14, 728 DEGs over 11,000 show a fold change of less than 2 fold. I would have doubts about the biological relevance of many of these. Similarly there are many peptides with extremely small fold changes in abundance.
4. All supplementary data sets need legends to make the data in them very clear. At the moment a reader has to guess units, and what each column means. These legends need to be clear and cover the content of each column in the data sets to enable easy re-use and understanding.
5. I think the number of total quantified transcripts is 32,548 not 549; Phosphosites – 26,949 not 26,950. I am not sure how the number of protein groups comes to 9035 (as stated in the text), the spreadsheet has 9508. Perhaps some groups are duplicates – again legends are needed to make these data understandable to others.
6. It would be useful to explain the rationale behind the network algorithm for biologists without them having to go back to other publications. E.g. what is dynamic time warping clustering? We understand clustering of similar profiles together but how does this vary? What is ICA? What are the underlying principles of RPT_STAR? This does not need to be provided in great detail but with enough information for a reader to understand the logic of the pipeline.
7. Figure 2A is really not very useful at all. It would be much better to have a figure that visualises the steps (i.e. what they do) rather than just naming them.

8. When making the first GRN, are the TFs used in the modelling just TFs that are DE in either transcript or protein abundance? And target genes are DE transcripts? And in the second network, are the DE phosphosites used in TFs? And DE transcripts are the potential targets? In both of these networks it would be useful to know how many regulators and targets were input.
9. I am not clear what clustering the gene-products means? Clustering the protein abundance? With the transcripts? Separately from transcripts? And what happens with these clusters? Is the cluster median used in the network inference? How do clusters inform the resulting network?
10. Much more information needs to be provided on how the networks from different time points differed, and how they were integrated.
11. Fig 2B is not very informative – it is difficult to see how much the networks overlap. Fig 2B does not show “cross-communication between the time points, where early regulators feed-forward into late regulators, and conversely where late regulators feed-back onto early regulators (Fig 2B).” This is interesting information and needs to be clearly explained and demonstrated.
12. In the kinase network – are only include increases in phosphorylation in kinases included as regulators? The text says phosphorylated kinases should be active “and thus useful”.
13. Were the target genes the same in all three networks? And how large were the networks? i.e. what proportion of components used as input were in the network model outputs.
14. How were the three networks merged? Just union?
15. The temporal flow of information through the network is not at all visible in Figure 3 in any informative way. There is no way to see the proportion of interactions at each time point. I suggest the authors find a better way of illustrating how the networks change over time. In a time analysis, how do the authors deal with edges that come up in multiple networks? Use the first time point? How many edges overlap in the different networks and crucially how many edges (or what proportion) overlap in a meaningful manner – i.e. with adjacent time points, rather than more randomly.
16. As mentioned in my summary, the case study of the phosphorylation sites did not need the network for prediction. One site was already known to be phosphorylated by BIN2 (S171) and the other two sites were also changing in intensity in the time series data. The network predicts BIN2 phosphorylating two of these sites (not the known one) but it also predicts other kinases doing this. How many other kinases are upstream of these sites in the network? The manuscript mentions 3 and others so the value of prediction is not clear.
17. The rationale for the selection of the three TFs for further investigation is not clear. There are MANY TFs in the top 35% so why those three?
18. The root cell type data needs clear information on where it is from and whether you have used lists of DEGs identified in the cited papers (ensure the relevant publications are cited in the figure legend too). I do not see how the cell type data contributes to a model whereby BR induces the cyclins via repression of BRON.

Minor points:

1. Fig S1 legend – make it clear that it is GO analysis of DE transcripts, proteins, sites – not just all things present at that time point. Highlight the BR GO terms in the figure?
2. “In agreement with our previous work in maize 45, we observed that the correlation between kinase protein abundance and kinase p-loop phosphorylation intensity (i.e. activation state) greatly varies depending on the time point (Fig S3)” This should be Fig S2
3. Lower case in *Nicotiana benthamiana*
4. When looking at the ChIP data, does directly bound mean binding anywhere around a gene? It would be useful to clarify the thresholds used.
5. “Thus, we were able to use our integrative omics network to identify previously unknown BIN2 phosphorylation sites on BES1 as well as validate its putative direct downstream targets identified by ChIP” The network didn’t validate the targets, the ChIP validated the network predictions.
6. The rationale for this is not clear to non-specialists “ Thus, we reasoned that TCX2 may regulate cell division in response to BR”
7. There is a repeated paragraph at the end of the ANL2 network motif section
8. Bron mutant characterisation – where is the location of the primers for gene expression analysis? Weaker impact on bron expression in bron-1 – depends on protein impact not expression
9. “undivided (i.e. actively dividing) CEI” – what are CEI?
10. Overlap of bron DEGs with BR DEGs – needs the source of these BR responsive genes made clear

REVIEWER COMMENTS

Reviewer #1 (Remarks to the Author):

Comment 1: The manuscript by Clark et al, reports an integrated omics study that profiled temporal changes in the abundance of transcript/protein and phosphorylation status using wild-type Arabidopsis seedlings that were pretreated with BRZ [a highly-specific inhibitor of brassinosteroid (BR) biosynthesis] followed with treatment with mock solution or 1 micromolar of brassinolide (BL, the most active member of the BR family) for different durations. The authors developed a network inference pipeline, named Spatiotemporal Clustering and Inference of Omics Network (SC-ION), to integrate data of their transcriptomic/proteomic/phosphoproteomic analyses of wild-type Arabidopsis seedlings treated with or without BL, creating a transcriptional factor(TF)-centered network and a kinase-centered network. Merging of the two networks revealed “the temporal signaling events of brassinosteroid response in Arabidopsis” with kinases at early events of the BR signaling followed by changes in abundance/phosphorylation status of TFs and subsequent transcriptional regulation of thousands of Arabidopsis genes. The BR signaling pathway is one of the best studied plant signaling processes. Previous extensive genetic, biochemical, and omics studies have established that BR signaling is executed by a protein phosphorylation-mediated biochemical cascade that regulates the abundance, subcellular locations, and biochemical activities of members of the BES1/BZR1 family. Loss-of-function mutations in BRI1, simultaneous elimination of all 6 BES1/BZR1 members, or gain-of-function mutations in BIN2 (an Arabidopsis homolog of the mammalian GSK3 kinases, which phosphorylates and negatively regulates BES1/BZR1) result in almost identical dwarf phenotypes. Thus, the current study merely confirmed what was already known for the BR signaling pathway for quite some time now.

Response 1: We agree that initial BR signaling events from perception through the phosphorylation cascade leading to the transcription factors BES1 and BZR1 is one of the better resolved signaling pathways in plants. However, our main focus is the broader BR responsive signaling network triggered by initial signaling components (BRI1, BIN2, BES1, BZR1 etc.). Our datasets provide a valuable resource for charting these networks given that they span both experimental time across BL treatment as well as different layers of regulation (transcript, protein and phosphorylation). We illustrate the utility of our multi-omics datasets through prediction and validation of the BRON subnetwork which involves both kinases and TFs.

Comment 2: One of the major concern that I have for this study is the lack of an important negative control, an Arabidopsis triple mutant lacking BRI1 and its two homologs, BRL1 and BRL2, which initiate BR signaling in Arabidopsis with BRI1 playing a predominant role and the other two being involved in vascular development. This control is extremely important to rid of phosphorylation sites (differentially expressed genes or changes in protein abundance) that might be caused by non-specific stress-response pathways (due to exogenous applications of BL at high concentrations). An earlier study (Zhang et al., 2009 PNAS 106:4543) showed that an aqueous buffer without any hormone could stimulate BES1 phosphorylation via an abscisic acid-mediated stress signaling pathway.

Response 2: We understand the reviewer’s concern about the experimental design. We first would like to note that we are comparing a mock control to a BL-treated sample at each time point. The mock control plants were submerged in an aqueous buffer in the same manner as the BL-treated samples.

Thus, by comparing the mock to the BL-treatment, any potential “stress” effects of the aqueous buffer are accounted for since both samples were handled in the same manner. In our GO analysis, we find many terms related to BR response, and we identify many DE gene-products that are known players in BR signaling, so we believe our methods are accurately recapitulating the BR response.

To reduce the amount of background BR signaling, all seedlings were pre-treated with 1 μ M BRZ for 7 days before the BL treatment. We checked the effect of BRZ pre-treatment and BL time course treatment by performing a Western Blot to monitor the phosphorylation status of BES1 protein as a read out of BL signaling (these updated BRZ blots which were also done on tissue used for profiling are shown Figure S1). We also performed Western Blots on BES1 in the *bri1-301*, *bri1-116*, *bri1-GABI*, and *bri1-116 bri1-2 bri1-3* mutants to show BES1 phosphorylation status in a null BR signaling background. We show the BRZ blots in Supplementary Figure 1 and include the full blot, including the *bri1* mutants, in the Source Data. From these experiments we see that BR signaling is mostly off in the mock treated samples and that signaling is induced following BL treatment. Thus, as detailed above, we are confident that our current experimental procedures allow us to accurately measure BR response and account for any stress response due to the aqueous buffer treatment.

Comment 3: The most interesting part of the study is discovery of BRONTOSAURUS (BRON) as a transcription factor that regulates cell division in Arabidopsis roots. Previous studies done in Ana Cano-Delgado's laboratory showed that BR controls meristem size and QC division through BES1. While the complete elimination of 6 members of the BES1/BZR1 family [BES1, BZR1, and 4 BES1 homologs (BEHs)] resulted in a morphological phenotype identical to that of a null *bri1* mutant, no study was done to

determine the impact of the sextuple *bes1/bzr1/beh* mutation on the root growth. It is possible that additional transcription factors, such as *BRON*, are involved in BR-regulated root growth via a *BES1/BZR1*-independent manner. The SC-ION analysis revealed that *BRON* is a target of *At1G06070* (also known as *bZIP69*, a member of the Arabidopsis *bZIP* transcription factor family), which is likely regulated by kinases known to be involved in early signaling events, such as *BAK1*, a coreceptor for *BRI1*, and two members of the BR signaling kinase (*BSK*) family involved in BR signaling from *BRI1/BAK1* to *BIN2*. Additional experiments are needed to fully establish the operation of the BR-activated kinase(s)-*bZIP69*-*BRON* cascade in BR-stimulated root growth. The presented results on *BRON* are too preliminary to support the existence of this cascade. Besides, the kinases that connected to *bZIP69* are also known to be involved in plant stress response, such as *SK41*, *BSK1*, *KIN10*, and *BAK1*.

Response 3: We agree with the reviewer that more work can be done to validate the *BRON* signaling cascade. To that end, we performed RT-qPCR to measure *BRON* expression in *kin10*, *mpk6*, *tmk1*, *bak1*, *map4ka1*, and *vdof2* mutants. We found that *BRON* expression is significantly altered in the *mpk6*, *tmk1*, *bak1*, *map4ka1*, and *vdof2* mutants. Additionally, we showed that *VDOF2* expression is significantly induced in *bron* mutants, supporting a negative feedback loop between *VDOF2* and *BRON*. These results have allowed us to expand our original model to incorporate how signaling through these kinases and *VDOF2* modulates *BRON* levels. All of these new results and the updated model are now presented in Figure 5, Supplementary Figure 5, Supplementary Table 3, and in lines 358-374.

We were unable to obtain confirmed homozygous mutants for the remaining predicted upstream regulators of *BRON*, including *bZIP69*. We ordered lines for *bsk1* and *bsk8* TDNA insertion mutants from the Arabidopsis Biological Resource Center, but the *bsk1* lines were heterozygous for the insertion and the *bsk8* line suffered from poor germination. However, we believe further pursuing these upstream regulators is beyond the scope of this manuscript.

In our current revision, we are careful to highlight that the regulation of *BRON* by these kinases and TF may or may not be direct and should be investigated further. However, we believe that our additional work on *BRON* better illustrates how it fits into the existing signaling cascade and provides a novel regulatory network that shows how BR may modulate the expression of cell cycle genes, like *CYCP4;1* and *CYCD3;1*, through the kinases *MAP4Ka1*, *BAK1*, *MPK6*, *TMK1*, and the TFs *VDOF2* and *BRON*.

Additional comments:

Comment 4: The authors should double check the numbers that were stated in the abstract. Are they absolutely sure that their transcriptome experiments profiled a total of 32,549 transcripts and their phosphor-proteomic assays detected a total of 26,950 phosphorylation sites? There are many genes with “1.39E-17” (meaning no detectable expression??) listed across all cells in Supplementary Table 1, and I think that those genes should be eliminated from their final transcript counts. There are many repetitive phosphopeptides with identical phosphorylated residues in Supplementary Table 3, and a careful analysis of these peptides are needed to get accurate count of detected phosphorylation sites (especially those with statistically significant changes IN RESPONSE to BL treatment, which should be included in a separate sheet of the excel file).

Response 4: We thank the reviewer for pointing this out. We have removed all gene-products with low expression across replicates and corrected the reported numbers. For clarity, we have now included the

original, data in separate sheets of the Datasets which were used to calculate the number of detected gene products. We have also included separate sheets for the DE gene products and a legend at the beginning of each file detailing what each sheet contains.

Comment 5: There are at least 16 additional BIN2 substrates (other than the 6 members of the BES1/BZR1 family) (Youn & Kim, 2015, Mol Plant 8:552-565), and BR treatment is expected to alter their phosphorylation status. The authors should discuss how many of these previously demonstrated BIN2 substrates were identified in their phosphoproteomic experiments given the large number (26,950) of detected phosphorylation sites. The immunoblot experiments shown in this manuscript and previously published studies suggested the presence of 20-25 putative BIN2 phosphorylation sites on BES1/BZR1. An earlier study (Ryu et al., 2010 Mol Cells 29:283) that utilized site-directed mutagenesis showed that the phosphorylation of Ser171 and Thr175 regulates the interaction of BES1 with 14-3-3 proteins for nuclear export while an earlier phosphoproteomic study (Nakagami et al., 2010 Plant Physiol 153:1161) detected at least 6 phosphorylation sites in BES1, including Ser169, Ser171, Ser179, Ser180, Thr182, and Ser183. The studies on Ser171 of BES1 and Ser173 of BZR1 revealed the important role of this conserved Ser residue in regulating the subcellular localization of BES1/BZR1, which is consistent with the detected phenotypes shown in Fig. S3E. A functional study that tests the role of Ser179/180 should be performed with a mutant BES1 construct carrying the Ser179Ser180-AA double mutation (instead of the Ser171Ser179Ser180-AAA triple mutation) in order to conclude that “Our study therefore identified three key BIN2 phosphorylation sites that negatively regulate BES1 activity and hence BR signaling” . The immunoblots shown in Fig. S3, C&D are not sufficient to conclude that Ser179/180 are BIN2 phosphorylation sites given that the phosphorylation experiments were performed with the tobacco transient expression system, which contains many kinases and phosphatases. An in vitro phosphorylation assay with purified BIN2 and BES1 (wild-type plus mutated ones) coupled with Phos-tag SDS-PAGE is needed to confirm that the two sites are indeed phosphorylated by BIN2 because the Ser179Ser180 sites do not conform to the S/TxxxS/T consensus phosphorylation motif of known GSK3 kinases.

Response 5: We agree that our previous results were not sufficient to conclusively distinguish the role of S171 phosphorylation from the S179/S180 sites we report as BIN2 phosphorylation sites. Thus, in this revision, we have generated BES1 with only S179 and S180 mutated to alanine (denoted as BES1^{S2A}). Using in vitro kinase assays coupled with phostag gel we found that the BES1^{S2A} protein was phosphorylated by BIN2 to a lesser degree compared to BES1, suggesting that S179 and S180 are true BIN2 phosphorylation sites.

We also generated transgenic Arabidopsis plants overexpressing BES1-FLAG or BES1^{S2A}-FLAG. The majority (20/26: 77%) of the BES1^{S2A}-FLAG T1 plants have longer leaf petiole and curly leaves, a characteristic of gain-of-function mutants in BR signaling. Conversely, only 1 of 14 (7%) BES1-FLAG plants grown at the same time exhibited these phenotypes. Together, our new results support our kinase-signaling network prediction that S179 and S180 are BIN2 phosphorylation sites that impact BES1 function. We have updated the text to reflect these changes (Lines 199-226), and our new results are shown in Supplementary Figure 3.

Comment 6: I am also curious if the phosphoproteomic experiments detected any phosphopeptide of BRI1, which was known to be auto/trans-phosphorylated at >30 sites. BRI1 is a quite abundant protein in Arabidopsis and the reported phospho-proteomic assays did detect phosphorylation sites of many

membrane-localized receptor-like kinases.

Response 6: We detected 8 different phosphosites of BRI1, which can be found in worksheet "MQ_unedited_Phospho (STY)Sites" of Supplementary Data 3. The IDs of these phosphosites are 16511-16515 and 25448-25450. All 8 of these sites are DE in response to BL in at least one time point (see worksheets Mock_vs_BR).

Reviewer #2 (Remarks to the Author):

Comment 7: The article entitled "Integrated omics networks reveal the temporal signaling events of brassinosteroid response in Arabidopsis" describes a systems biology approach to integrate multi-omic datasets and unravel the molecular signalling events of BR response in Arabidopsis. The BR signalling pathways are very well studied in Arabidopsis, however having a good integrative network analysis applied to multi-omics data in a signalling pathway is new. The authors combined in this paper not only large scale bioinformatics analysis but also showed functional data for a few newly identified genes in the BR signalling. The paper is well written and clear.

Response 7: We thank the reviewer for their positive comments on our manuscript.

I have minor comments to improve the content of the paper.

Comment 8: - What I missed in the paper is the transactivation (early signaling cascade in the BR signaling) of the kinase domains of the receptors involved in the BR perception for example BRI1, BAK1 and others.

Response 8: Please see Response 6 for information on BRI1 phosphosites. We also detected 8 different phosphosites of BAK1, which can be found in worksheet "MQ_unedited_Phospho (STY)Sites" of Supplementary Data 3. The IDs of these phosphosites are 15930-15932, 25301-25304, and 26851. Sites 15932, 25301, 25302, 25303, and 26851 (5 out of 8 sites) are DE in response to BL in at least one time point (see worksheets Mock_vs_BR).

While identifying DE phosphosites in known kinases involved in BR response is interesting, it is not our main focus of the paper. Rather, we used our multi-omics data to identify novel, downstream players in the BR signaling cascade such as BRONTOSAURUS.

Comment 9: - I would like to see more explanation on the Network patterns observed and on conserved network regulations (can you explain using these feedforward and feedback loops how the application of BL will act on the different TFs in the BL transcription network to regulate the signalling response? You have the data so use these networks to describe the interaction and signaling (not just provide a figure with multiple interactions but discuss the interactions and the patterns you see). For example can you make predictions based on the network motifs you observed about how the signaling will change if one TF (for example BRON is not functional)? What will happen to the network observed, can you give an example in combination with your functional data for BRON (including RNA-seq of the mutant).

Response 9: We thank the reviewer for this great point. In our revised manuscript, we have done additional work on validating the upstream regulators of downstream targets of BRON. This has allowed us to identify the network motifs that are important for BRON's regulatory network, specifically the negative feedback loop between BRON and VDOF2. We further discuss the potential impacts of this

negative feedback loop on BRON expression in lines 464-469.

Reviewer #3 (Remarks to the Author):

Comment 10: This manuscript describes RNAseq and TMT labelled phosphoproteomics revealed the molecular signaling events of brassinosteroids(BR) response in Arabidopsis. In this study, total 32,549 transcripts, 9,035 protein groups, and 26,950 phosphorylation sites from Arabidopsis seedlings treated with brassinolide for six different lengths of time were integrated by a network inference pipeline called Spatiotemporal Clustering and Inference of Omics Networks (SC-ION). Phosphorylation sites on BES1 were experimentally validated their importance. For further study, functional studies of BRONTOSAURUS (BRON) as a transcription factor that regulates cell division were performed. Overall, proteomics studies were well performed and find interesting clues for phosphorylation site of BES in BR signaling.

Response 10: We thank the reviewer for their enthusiasm for our manuscript.

Comment 11. Authors used 10plex TMT kit for labelling of tryptic digested peptides, but Arabidopsis seedlings treated with brassinolide for six different lengths of time were used for proteomics study. Authors should provide more detailed TMT labelling condition and need pooling sets for normalization of quantification.

Response 11: We agree a clear labeling description is important. The TMT-labeling strategy is provided in what is now Supplementary Table 5. We describe in Methods that we ran a pooled reference in two TMT channels. The pooled reference was comprised of equal amounts of peptides take from each sample and mixed together (lines 540-541). The pooled reference was split in two and then labeled with either TMT_126 or TMT_131.

Comment 12. Tryptic digestion and TMT labelling efficiency are required.

Response 12: The trypsin digestion efficiency was 76.2, 21.7, and 2.1% for 0, 2, and 2 missed cleavages, respectively.

We added the statement that “Labeling efficiency was checked by performing a 60-minute reverse-phase run on 200 ng of TMT-labeled peptides. All TMT sets had labeling efficiencies \geq 95% (Supplementary Table 5).” to the Methods (lines 546-548) and now list the exact labeling efficiency for each TMT set in Supplementary Table 5.

Comment 13. LC-MSMS but not LC/MS-MS (line 445)

Response 13: Thank you, we have fixed this typo.

Comment 14. I suggest that there are a weak correlation between proteomic studied and functional studies of BRONTOSAURUS (BRON).

Response 14: We have included much more functional work on BRON in this revision, mainly RT-qPCR validation of regulation of BRON by multiple BR-responsive kinases and the transcription factor VDOF2. (See Response Summary and Response 3 for more details).

Reviewer #4 (Remarks to the Author):

This paper outlines the generation of time series data on Arabidopsis transcript, protein and peptide phosphorylation in response to brassinosteroid treatment of seedlings. This data is then used in network inference to generate networks predicting the impact of transcription factor abundance (protein or transcript) on transcript levels of downstream target genes, or to predict the impact of peptide phosphorylation at specific sites on downstream transcript abundance. From this network the authors identify two phosphorylation sites in BES1 phosphorylated by BIN2; the role of ANL2 in root growth inhibition by BR; a similar role for BRON with additional impacts on root cell division; and profile gene expression in bron mutant roots compared to wildtype. The incorporation of time series peptide phosphorylation data with transcript/protein abundance is an interesting approach and provides plenty of predictions for experimental testing. However, unfortunately the prediction of the phosphorylation of BES1 by BIN2 could have been made from the phosphorylation data alone, given the known role of BIN2 in phosphorylating another site in BES1 already. The choice of transcription factors for experimental testing is not explained and the findings on the role of ANL2 are minimal. BRON is investigated in slightly more detail but the gene expression profiling does not yield obvious potential mechanisms (especially given the lack of any cell cycle gene enrichment). The manuscript is well written in places but considerable details are missing, and key findings (e.g. around timing of BR response) are not at all clear from the current presentation and explanation. In summary, the data sets are valuable and will no doubt be of value to the community but evidence for the relevance and importance of the network inference is not strong, and the current manuscript does not provide significant advance in our understanding of BR signaling and response. I have provided further information below.

Major points:

Comment 15. I could not see any information on the normalisation process for the protein data. Are internal references used? Whole proteome methods?

Response 15: A description of how the pooled references used for normalizing the protein data can be found in the “Tandem Mass Tag (TMT) Labeling” section of the Methods (Lines 540-541). Normalization is described in the “Proteomics Data Analysis” of the Methods (lines 614-616).

Comment 16. The identification of differentially expressed genes is not clear. Do the authors use the time series nature of the data set? Or simply do pairwise comparisons at each time point? I think it is the latter in which case it would be useful to utilise the time series design to refine these.

Response 16: Benchmark studies have shown that pairwise comparisons at each time point outperform dedicated time series analysis tools when experiments contain 8 or fewer time points (Spies et al 2019, Brief. Bioinform.). Therefore, we opted to use a set of pairwise comparisons for our DE analysis. Since we always compare to the mock samples at each time point we are focusing on the effect of BL treatment rather than changes due to differences in the age of the plant, circadian rhythms, or other factors. We have added statements to the Methods section to make clear that all DE comparisons are comparing the mock to the BL-treated samples at each individual time point (lines 520-522, 619-621).

Comment 17. The list of differentially expressed genes is not particularly rigorous. It would be useful to have information on how many DEGs were specific to a single time point - and greater confidence could be placed in DEGs that were DE over multiple time points. In addition, no fold change threshold has been

used. Out of 14, 728 DEGs over 11,000 show a fold change of less than 2 fold. I would have doubts about the biological relevance of many of these. Similarly there are many peptides with extremely small fold changes in abundance.

Response 17: We have included information on how many differentially expressed gene products are specific to each time point in Figure 1c and in the Supplementary Information. We did use a fold change threshold of 1.25 for the transcriptomic data and of 1.1 for the (phospho)proteomics data, which is stated in the Methods (lines 522 and 621). A fold-change cutoff is inherently arbitrary as there is no minimum fold-change that is biologically relevant and likely will be gene-specific. We provide the full dataset so that readers can select a fold-change cutoff specific for their experimental goals.

Comment 18. All supplementary data sets need legends to make the data in them very clear. At the moment a reader has to guess units, and what each column means. These legends need to be clear and cover the content of each column in the data sets to enable easy re-use and understanding.

Response 18: We agree and we have added a legend to the first sheet of every supplemental dataset to clarify units, column names, etc.

Comment 19: I think the number of total quantified transcripts is 32,548 not 549; Phosphosites - 26,949 not 26,950. I am not sure how the number of protein groups comes to 9035 (as stated in the text), the spreadsheet has 9508. Perhaps some groups are duplicates - again legends are needed to make these data understandable to others.

Response 19: Thank you for pointing this out. We have fixed this in the revised manuscript.

Comment 20: It would be useful to explain the rationale behind the network algorithm for biologists without them having to go back to other publications. E.g. what is dynamic time warping clustering? We understand clustering of similar profiles together but how does this vary? What is ICA? What are the underlying principles of RPT_STAR? This does not need to be provided in great detail but with enough information for a reader to understand the logic of the pipeline.

Response 20: We agree and we have now included a Supplemental Methods section in the Supplementary Information that includes more details on the RTP-STAR, GENIE3, DTW, and ICA components of the pipeline.

Comment 21: Figure 2A is really not very useful at all. It would be much better to have a figure that visualises the steps (i.e. what they do) rather than just naming them.

Response 21: We have kept Figure 2a as it provides a general summary for how the pipeline works. However, we have added Supplementary Figure 7 which provides a better visualization of how the network pipeline works. We also include more detail in the Supplementary Methods section (see Supplementary Information).

Comment 22: When making the first GRN, are the TFs used in the modelling just TFs that are DE in either transcript or protein abundance? And target genes are DE transcripts? And in the second network, are the DE phosphosites used in TFs? And DE transcripts are the potential targets? In both of these networks it would be useful to know how many regulators and targets were input.

Response 22: We apologize for any confusion our methods have caused. We have expanded our methods section to provide more detail on the network construction:

“Two TF-centered networks were inferred (Supplementary Figure 7). In the first network, TF protein abundance (when quantified) or TF transcript abundance (when cognate protein was not quantified) was used as the “regulator” value to infer their “target” genes’ transcript abundance. In the second phosphosite network, we inferred the transcript levels of “target” genes using TF phosphosite intensities as “regulators.” In both the networks, only the regulators and targets DE at each time point were used in these individual subnetworks. Next, the clustering matrix was constructed by combining the “regulator” (TF abundance) and “target” (transcript levels) data so that genes were clustered based on the protein/phosphosite levels of the regulators and transcript levels of the targets. So, for example, TF protein abundance is clustered with the transcript abundance of its potential target genes. Clustering was then performed using the DTW method given the temporal nature of our data. After clustering, six sub-networks were inferred for each TF-centered network, where each sub-network represents the regulations predicted to happen using only one time point (15 min, 30 min, 1 hr, 2 hr, 4 hr, 8 hr). This allows us to denote which regulations are predicted to happen early or late in BR response. Further, TFs are only predicted to regulate transcripts in their same cluster. This allows the clustering to inform the network prediction by reducing the possible number of edges. The six subnetworks were combined in a union in Cytoscape to form the final abundance and phosphosite networks. Cytoscape was also used to create the merged abundance, phosphosite, and kinase signaling networks (Supplementary Data 5).” (Lines 643-664).

We now include the numbers of targets, regulators, and edges for each inferred network in Dataset S5 so that they may be clearly accessed by all readers.

Comment 23: *I am not clear what clustering the gene-products means? Clustering the protein abundance? With the transcripts? Separately from transcripts? And what happens with these clusters? Is the cluster median used in the network inference? How do clusters inform the resulting network?*

Response 23: Please see Response 22 for our text from the main methods on the clustering. We have added additional details on the clustering methods in the Supplementary Methods (see Supplementary Information).

Comment 24: *Much more information needs to be provided on how the networks from different time points differed, and how they were integrated.*

Response 24: Please see Response 22 for our text from the main methods on how the networks are combined from different time points.

Comment 25: *Fig 2B is not very informative - it is difficult to see how much the networks overlap. Fig 2B does not show “cross-communication between the time points, where early regulators feed-forward into late regulators, and conversely where late regulators feed-back onto early regulators (Fig 2B).” This is interesting information and needs to be clearly explained and demonstrated.*

Response 25: We agree that visualization of large networks is difficult. We maintain that the network visualization in Figure 2b provides an overall picture and broadly shows the cross communication between time points. The colors of the edge represent the time point. When viewing the network, one can see that the time-points are arranged in order from the top of the circle, starting with 15 minutes and working clockwise to 8hr. One can also see that the different colored lines connect different parts of the circle. For example, some of the dark blue (8hr edges) connect to the top of the circle, which is where the genes DE at 15 minutes are.

Comment 26: *In the kinase network - are only include increases in phosphorylation in kinases included as regulators? The text says phosphorylated kinases should be active "and thus useful" .*

Response 26: Sorry for the confusion with the text in the Results section. We include kinases with both decreased and increased activation loop phosphorylation as both inactivation and activation of a kinase is useful for prediction. Lines 168-173 in the Results now read:

"In this network, we considered kinases with DE phosphosites in their p-loop (also termed activation loop; A-loop) domain as potential regulators, as phosphorylation in this region is necessary for kinase activity⁴¹. Thus, activation loop phosphorylation can be used to infer kinase activity and is useful for predicting kinase-signaling^{40,42,43}".

In the Methods we also state "Kinases with phosphosites in the activation loop, also called p-loop, domain that were DE in response to BL were used as the potential regulators. All genes with phosphosites that were DE in response to BL were used as the potential targets." (lines 672-674) We also now provide a visualization of the workflow in Supplementary Figure 7.

Comment 26: *Were the target genes the same in all three networks? And how large were the networks? i.e. what proportion of components used as input were in the network model outputs.*

Response 26: We now include the number of regulators, targets, and edges in each network in Dataset S5. The target genes were not the same in all three networks. In the TF-abundance and TF-phosphosite networks, DE transcripts were used as targets. In the kinase-signaling network, DE phosphosites were used as targets. We hope that the updated methods we have written in Response 22 and Response 26 help to reduce any confusion about the target genes.

Comment 27: *How were the three networks merged? Just union?*

Response 27: Yes, the networks were merged by union as described in the Methods: "The six subnetworks were combined in a union in Cytoscape to form the final abundance and phosphosite networks." (lines 661-662).

Comment 28: *The temporal flow of information through the network is not at all visible in Figure 3 in any informative way. There is no way to see the proportion of interactions at each time point. I suggest the authors find a better way of illustrating how the networks change over time. In a time analysis, how do the authors deal with edges that come up in multiple networks? Use the first time point? How many edges overlap in the different networks and crucially how many edges (or what proportion) overlap in a meaningful manner - i.e. with adjacent time points, rather than more randomly.*

Response 28: It is difficult to fully visualize these integrated multi-omics networks due to the large amount of information they contain. We do not currently have a good way of visualizing both time-point-specific and gene-product-specific components in the same image. Thus, we chose to prioritize time-point-specific visualization in Figure 2 and gene-product-specific visualization in Figure 3 so that readers can see both.

As stated in Response 24, the networks are combined in a union, so edges that appear in multiple networks are kept. While the questions about the number of overlapping edges are interesting, we believe they are beyond the scope of this paper. Our main goal with the network analysis is to use our NMS to identify new genes of interest, such as BRON. We do include all of the edges inferred and

detailed information on each edge, such as the time point and gene-product used, in Supplementary Data 5. This dataset could be used to answer intriguing questions such as these in the future.

Comment 29: As mentioned in my summary, the case study of the phosphorylation sites did not need the network for prediction. One site was already known to be phosphorylated by BIN2 (S171) and the other two sites were also changing in intensity in the time series data. The network predicts BIN2 phosphorylating two of these sites (not the known one) but it also predicts other kinases doing this. How many other kinases are upstream of these sites in the network? The manuscript mentions 3 and others so the value of prediction is not clear.

Response 29: Other reviewers also pointed out that the inclusion of S171 was not supported by the network and prevents us from showing that S179/S180 alone are BIN2-phosphorylated sites. Thus, in this revision, we have generated BES1 with only S179 and S180 mutated (denoted as BES1^{S2A}). Please see Response Summary and Response 5 for more details.

Comment 30: The rationale for the selection of the three TFs for further investigation is not clear. There are MANY TFs in the top 35% so why those three?

Response 30: We chose these three TFs as we were able to obtain confirmed homozygous mutants for functional validation. Additionally, none of these three TFs have known roles in BR response.

Comment 31: The root cell type data needs clear information on where it is from and whether you have used lists of DEGs identified in the cited papers (ensure the relevant publications are cited in the figure legend too). I do not see how the cell type data contributes to a model whereby BR induces the cyclins via repression of BRON.

Response 31: We apologize for the confusion. These data were obtained from previously published papers (ref #s 35 and 51). We took the reported FPKM expression data for these genes from each of these papers and normalized the values to the mean expression of each gene before plotting in Supplementary Figure 6. We now make sure it is clear in the figure legend where these data come from, and how they are normalized. We refer the reviewer to these studies for the experimental details.

The cell type specific data show that these cyclins and *BRON* are co-expressed spatially. We show in the BR time course that BR induces cyclins, and we show that these cyclins are induced in the *bron* mutant. This leads us to propose that BR regulates these cyclins through *BRON*.

Minor points:

Comment 32: Fig S1 legend - make it clear that it is GO analysis of DE transcripts, proteins, sites - not just all things present at that time point. Highlight the BR GO terms in the figure?

Response 32: We have clarified this in the figure legend and have highlighted the suggested terms. Thank you for the suggestion.

Comment 33: "In agreement with our previous work in maize 45, we observed that the correlation between kinase protein abundance and kinase p-loop phosphorylation intensity (i.e. activation state) greatly varies depending on the time point (Fig S3) " This should be Fig S2

Response 33: Thank you, we have made sure the correct figure (now Supplementary Figure 2) is cited in this statement (line 175).

Comment 34: Lower case in *Nicotiana benthamiana*

Response 34: Thank you, we have fixed this typo.

Comment 35: When looking at the ChIP data, does directly bound mean binding anywhere around a gene? It would be useful to clarify the thresholds used.

Response 35: We have added this statement to the methods:

“Targets in the first-neighbor BES1 subnetwork (Fig 3) were validated using a list of targets shown to be bound by BES1 or its homologue BZR1 in various ChIP studies¹³⁻¹⁵. The results of this validation are reported in Table S1” (Lines 689-691). For more details on the ChIP experiments, we refer the reviewer to the published studies that these ChIP data were obtained from (ref #s 13-15).

Comment 36: *“Thus, we were able to use our integrative omics network to identify previously unknown BIN2 phosphorylation sites on BES1 as well as validate its putative direct downstream targets identified by ChIP”* The network didn't validate the targets, the ChIP validated the network predictions.

Response 36: This is a good point, thank you. We have flipped this sentence.

Comment 37: The rationale for this is not clear to non-specialists *“Thus, we reasoned that TCX2 may regulate cell division in response to BR”*

Response 37: Great point and sorry for the lack of clarity. We have added the sentence

“Additionally, TCX2 has a documented root stem cell division phenotype⁴⁰” to lines 288-289. TCX2 has a published cell division phenotype and is responsive to BR in our time course. Thus, we reason the two may be connected.

Comment 38: There is a repeated paragraph at the end of the ANL2 network motif section

Response 38: Thank you, we have deleted the repeated paragraph.

Comment 39: *Bron mutant characterisation - where is the location of the primers for gene expression analysis? Weaker impact on bron expression in bron-1 - depends on protein impact not expression*

Response 39: The primers span the last exon-exon junction. We checked the specificity of the primers using a melt curve. We also have repeated our original RT-qPCR on the *bron* mutants and obtained similar results. These results are now shown in Figure 5, Supplementary Figure 5 and Supplementary Table 3.

Comment 40: *“undivided (i.e. actively dividing) CEI” - what are CEI?*

Response 40: Great catch. We now define QC (Quiescent Center) and CEI (Cortex Endodermis Initials) in the manuscript in lines 299 and 321.

Comment 41: *Overlap of bron DEGs with BR DEGs - needs the source of these BR responsive genes made clear*

Response 41: The BR DEGs are those identified in the time course. We have clarified this in line 338.

REVIEWERS' COMMENTS

Reviewer #2 (Remarks to the Author):

I think the authors did a great job to improve the paper. Importantly additional experiments were performed to clarify the identified network interactions. For example functional work on BRON illustrates a novel regulatory network that shows how BR may modulate the expression of cell cycle genes, like CYCP4;1 and CYCD3;1 in the root, through the kinases MAP4Ka1, BAK1, MPK6, TMK1, and the TFs VDOF2 and BRON.

Reviewer #3 (Remarks to the Author):

Comments were well addressed and improved revised manuscript.

No comments for revised manuscript now.

Reviewer #4 (Remarks to the Author):

The authors have addressed the majority of my concerns and also provided a very clear point by point response.

The additional information and supplementary figure on network inference is very helpful.

For Figure 2b – I understand better given the author's explanation. They seem to suggest that the nodes in the network are arranged in the circle in order of timing (genes DE at 15 mins at the top) so then it is obvious that some late edges impact early nodes. I suggest that this information is provided in the figure legend – at the moment there is minimal information on what the circles indicate.

REVIEWER REQUESTS

Reviewer #4 (Remarks to the Author): The authors have addressed the majority of my concerns and also provided a very clear point by point response.

The additional information and supplementary figure on network inference is very helpful.

For Figure 2b –I understand better given the author’s explanation. They seem to suggest that the nodes in the network are arranged in the circle in order of timing (genes DE at 15 mins at the top) so then it is obvious that some late edges impact early nodes. I suggest that this information is provided in the figure legend –at the moment there is minimal information on what the circles indicate.

We have added the additional statement to the figure legend for Figure 2b: “Nodes represent individual genes. The nodes are arranged in a circular layout relative to timing – genes differentially expressed at the earliest time point (15 minutes) are placed at the top of the circle, and time increases as one moves clockwise through the layout.”